# Global translation during early development depends on the essential transcription factor PRDM10

Brenda Y. Han [1], Michelle K. Y. Seah[1], Imogen R. Brooks [1], Delia H. P. Quek[1], Dominic R. Huxley [1], Chuan-Sheng Foo [2], Li Ting Lee [1], Heike Wollmann[1], Huili Guo [1,3], Daniel M. Messerschmidt [1✉] & Ernesto Guccione [1,4✉]

Members of the PR/SET domain-containing (PRDM) family of zinc finger transcriptional regulators play diverse developmental roles. PRDM10 is a yet uncharacterized family member, and its function in vivo is unknown. Here, we report an essential requirement for PRDM10 in pre-implantation embryos and embryonic stem cells (mESCs), where loss of PRDM10 results in severe cell growth inhibition. Detailed genomic and biochemical analyses reveal that PRDM10 functions as a sequence-specific transcription factor. We identify *Eif3b*, which encodes a core component of the eukaryotic translation initiation factor 3 (eIF3) complex, as a key downstream target, and demonstrate that growth inhibition in PRDM10-deficient mESCs is in part mediated through EIF3B-dependent effects on global translation. Our work elucidates the molecular function of PRDM10 in maintaining global translation, establishes its essential role in early embryonic development and mESC homeostasis, and offers insights into the functional repertoire of PRDMs as well as the transcriptional mechanisms regulating translation.

[1] Institute of Molecular and Cell Biology (IMCB), Agency for Science, Technology and Research (A*STAR), Singapore, Singapore. [2] Institute for Infocomm Research (I2R), Agency for Science, Technology and Research (A*STAR), Singapore, Singapore. [3] Department of Biological Sciences, National University of Singapore, Singapore, Singapore. [4] Mount Sinai Center for Therapeutics Discovery, Departments of Pharmacological Sciences and Oncological Sciences, Tisch Cancer Institute, Icahn School of Medicine at Mount Sinai, New York, NY 10029, USA. ✉email: danielm@imcb.a-star.edu.sg; ernesto.guccione@mssm.edu

PRDM proteins are characterized by the presence of a conserved N-terminal PR (PRDI-BF1 and RIZ1) homology domain closely related to the lysine methyltransferase SET domain, followed by variable C2H2-type zinc finger repeats that typically mediate sequence-specific DNA binding. Several PRDMs have been shown to act as important transcriptional regulators controlling cell fate specification in various developmental contexts[1–3]. For example, Prdm1 is required for primordial germ cell specification and branchial arch patterning during embryonic development[4], and also plays an important role in regulating hematopoietic lineage differentiation[5]. Prdm16 promotes brown fat adipogenesis[6,7] and hematopoietic stem cell maintenance[3]. We and others have uncovered a critical and nonredundant role for Prdm14 and Prdm15 in maintaining naïve pluripotency of embryonic stem cells[8,9].

Prdm10, also known as tristanin[10], is highly conserved in vertebrates and belongs to the same phylogenetic subfamily as Prdm15[11]. Prdm10 is expressed in various embryonic and adult tissues[12,13]. A large-scale phenotypic screen revealed that homozygous deletion of Prdm10 in mice is embryonic lethal[14], and gene rearrangements involving PRDM10 have been described in some undifferentiated pleomorphic sarcomas[15,16]. Despite its potential biological significance, the molecular and functional properties of PRDM10 remain largely unknown, and its role in vivo has not been well-characterized.

In this study, we establish a conditional Prdm10 knockout mouse model to uncover a critical role for PRDM10 during very early embryonic development, and utilize mouse embryonic stem cells (mESCs) to study PRDM10's biochemical and molecular properties. We demonstrate that PRDM10 acts as a transcription factor that binds to the promoters of target genes and regulates their expression. Through direct transcriptional regulation of Eif3b, a key translation initiation factor, we show that PRDM10 plays a critical role in maintaining global translation essential for mESC survival.

## Results

### PRDM10 is essential for preimplantation embryogenesis.
Prdm10 encodes a protein containing an N-terminal PR domain, followed by ten C2H2 zinc fingers and a C-terminal glutamine (Q)-rich transactivation domain (Supplementary Fig. 1a) which is unique among the 17 PRDM family members. To explore the function of PRDM10 in vivo, we generated mice bearing a conditional allele (Prdm10$^F$) in which exon 5 of Prdm10 is flanked by loxP sites (Supplementary Fig. 1b). Cre-mediated removal of exon 5 introduces a frameshift resulting in a nonfunctional truncated protein (Supplementary Fig. 1a), thus generating a null allele (Prdm10$^Δ$) (Supplementary Fig. 1b). While Prdm10$^{Δ/+}$ mice were viable and fertile with no gross morphological or behavioral abnormalities observed in daily husbandry, no Prdm10$^{Δ/Δ}$ live pups were recovered from heterozygous intercrosses.

This prompted us to examine embryos from Prdm10$^{Δ/+}$ intercrosses at pre- and post-implantation time-points to define the timing of embryonic lethality. While at embryonic day (E) 3.5, all genotypes were recovered at close to Mendelian ratios, Prdm10$^{Δ/Δ}$ embryos were slightly underrepresented at E4.5, and no viable Prdm10$^{Δ/Δ}$ embryos were recovered at E7.5 and E12.5 postimplantation stages (Fig. 1a, Supplementary Fig. 1c). At E3.5, Prdm10$^{Δ/Δ}$ embryos had an abnormal, morula-like appearance (86%), in contrast to control embryos being mostly expanding/ expanded blastocysts (Fig. 1b, c and Supplementary Fig. 1d). Consistently, ex vivo cultured Prdm10$^{Δ/Δ}$ embryos developed normally from 2-cell to morula stage yet failed to form expanded blastocysts (Supplementary Fig. 1e). These data are consistent with the timing of embryonic death observed in utero and

furthermore show that lethality is due to an embryo-intrinsic defect independent of implantation failure. Given these observations and the complete absence of Prdm10$^{Δ/Δ}$ embryos at postimplantation stages, we conclude that loss of PRDM10 causes developmental arrest before blastocyst formation and embryonic lethality peri-implantation.

Remarkably, despite the fully penetrant preimplantation stage lethality phenotype, with evidence of increased cell apoptosis (Supplementary Fig. 2a), Prdm10$^{Δ/Δ}$ embryos still expressed lineage-specific markers such as OCT4 (inner cell mass; ICM), CDX2 (trophectoderm; TE), and NANOG (epiblast) at detectable levels (Supplementary Fig. 2a–c). This suggests that PRDM10 is not required for inducing lineage segregation, even though it is essential for embryo survival and developmental progression beyond preimplantation stages.

### Prdm10-null mESCs show reduced growth and increased apoptosis.
To facilitate the investigation of PRDM10's cellular and molecular functions in early development, we employed the strategy of using mESCs as an in vitro model, thus circumventing accessibility limitations to the embryo. We generated Prdm10$^{F/F}$; ROSA26-CreER$^{T2}$ mESCs in which Cre-mediated deletion is inducible by 4-hydroxytamoxifen (4-OHT) exposure. Recombination efficiency upon induction was verified at the genomic (Supplementary Fig. 3a), transcript (Fig. 1d, Supplementary Fig. 3b) and protein level (Fig. 1e, Supplementary Fig. 3c). Phenotypic characterization was performed on Prdm10$^{F/F}$; ROSA26-CreER$^{T2}$ mESCs maintained in serum-containing medium with leukemia inhibitory factor (serum/LIF).

Following acute deletion of Prdm10, we observed a reduction in cell growth rates becoming detectable starting around 3–4 days post-deletion and increasing in severity thereafter (Fig. 1f). Consistent with the observed growth defect, Prdm10-null mESCs formed smaller colonies that expanded poorly compared with controls (Fig. 1g, h), although colony morphology was not significantly altered (Fig. 1g). Crucially, re-introduction of full-length Prdm10 (Supplementary Fig. 3d, e) was sufficient to rescue the growth defects in Prdm10$^{Δ/Δ}$ cells (Supplementary Fig. 3f). Also of importance, Prdm10-null mESCs cultured in defined serum-free 2i medium with LIF (2i/LIF) showed a similar defect in growth (Supplementary Fig. 3g), suggesting that the phenotypic consequences of PRDM10 deficiency are not rescued by MEK or GSK3 inhibition, and more generally, that the requirement for PRDM10 in mESCs is independent of specific culture conditions.

To understand the basis for impaired cell growth in Prdm10-deficient mESCs, we evaluated possible impacts on cell survival and proliferation. For one, the loss of PRDM10 had no significant effect on cell cycle distribution (Supplementary Fig. 3h), suggesting that cell cycle progression is unaffected in Prdm10$^{Δ/Δ}$ mESCs. However, Prdm10$^{Δ/Δ}$ mESCs exhibited significantly higher levels of caspase 3/7 activity compared with controls (Fig. 1i), particularly at later time-points post-recombination (from Day 5 onwards), consistent with the kinetics of phenotypic onset observed in cell growth assays (Fig. 1f). These data demonstrate that PRDM10 deficiency leads to significantly increased apoptotic cell death in mESCs, and suggest that the Prdm10-null phenotype is independent of proliferation defects but instead can be mostly attributed to decreased cell survival.

### PRDM10 is dispensable for pluripotency and differentiation.
Previous studies from our group and others have identified two essential members of the PRDM family, PRDM14 and PRDM15, as key regulators of naïve pluripotency in ESCs[8,9,17], which raises the question of whether PRDM10 may also be required for maintenance of mESC pluripotency. To address this, we assessed

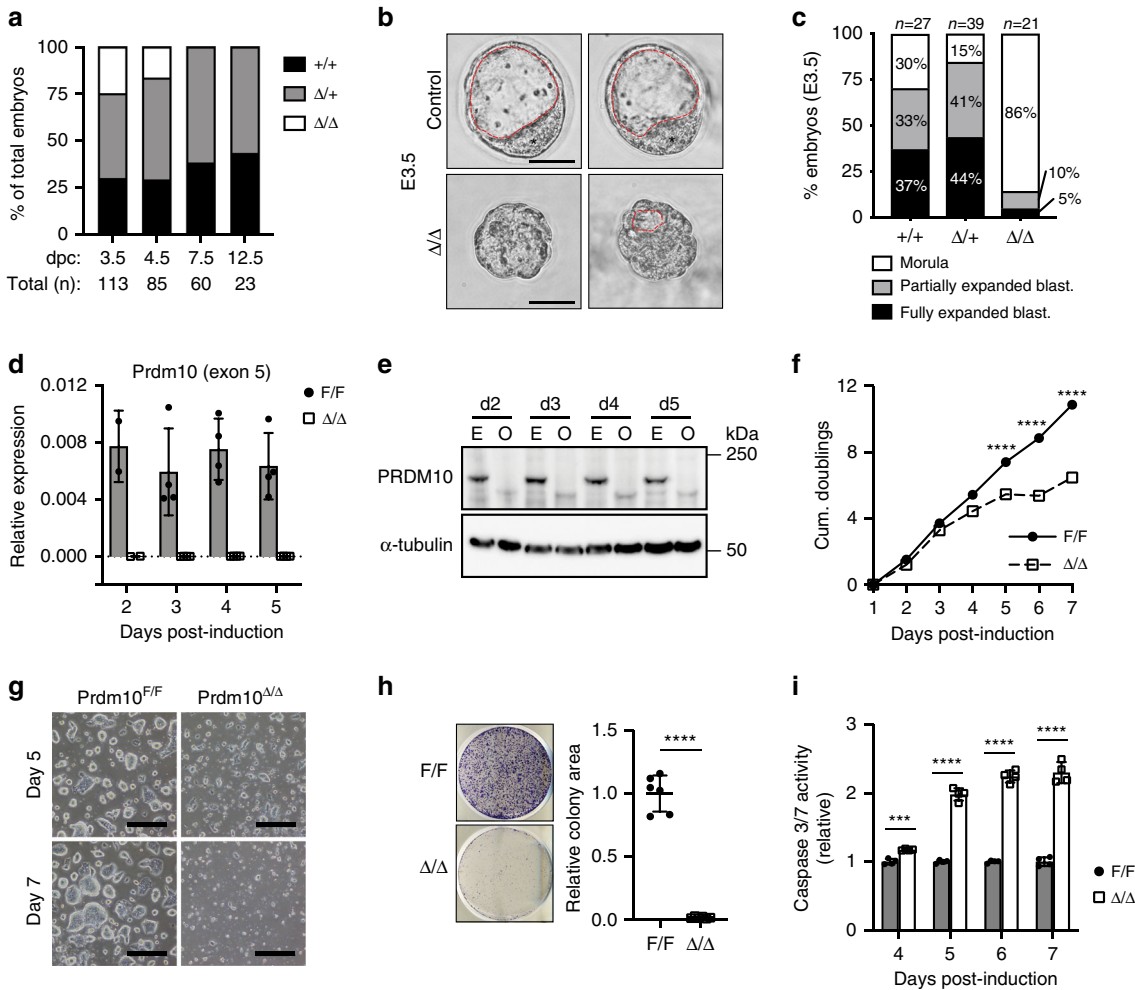

**Fig. 1 Prdm10 is essential for mouse preimplantation embryogenesis and mESC growth. a** Frequency of embryo genotypes obtained from heterozygous intercrosses at each developmental stage. E3.5 embryos are recovered at the expected Mendelian distribution; no $Prdm10^{\Delta/\Delta}$ embryos are observed by E7.5. **b** Representative images of mutant ($Prdm10^{\Delta/\Delta}$) and control ($Prdm10^{+/+}$, $Prdm10^{\Delta/+}$) embryos isolated at E3.5. The inner cell mass (ICM) is labeled with an asterisk, and the blastocoel is defined by a red dashed line. Scale bar: 50 μm. **c** Scoring of E3.5 embryos into three phenotypic categories: morula, partially expanded blastocyst, or fully expanded/cavitated blastocyst. $n = 27$ ($Prdm10^{+/+}$), $n = 39$ ($Prdm10^{\Delta/+}$), $n = 21$ ($Prdm10^{\Delta/\Delta}$). **d** qRT-PCR analysis of $Prdm10$ exon 5 expression in OHT-treated $Prdm10^{F/F}$; $CreER^{T2}$ ($\Delta/\Delta$) mESCs compared with vehicle-treated (F/F) controls at indicated time-points post-induction. Expression normalized to $Ubb$; $n = 3$ biological replicates. **e** Western blot analysis of PRDM10 protein levels in $Prdm10^{F/F}$; $CreER^{T2}$ mESCs at indicated time-points (days) after exposure to EtOH (E) or OHT (O). Loading control, α-tubulin. **f** PRDM10-depleted mESCs exhibit an increasingly severe cell growth defect over time. Cells were passaged at constant density every 2 days and counted daily up to Day 7 post-induction; $n = 4$ samples. Y-axis: cumulative population doublings. **g** Representative brightfield images of $Prdm10^{F/F}$ and $Prdm10^{\Delta/\Delta}$ mESC colonies at Day 5 and 7 post-induction. Cells were plated at equal densities 2 days prior to image acquisition. Scale bar: 500 μm. **h** Representative images from colony formation assay; $n = 6$. $Prdm10^{F/F}$ and $Prdm10^{\Delta/\Delta}$ mESCs were seeded at Day 3 post-induction and fixed for analysis at Day 8. **i** Caspase 3/7 activity in $Prdm10^{\Delta/\Delta}$ mESCs relative to $Prdm10^{F/F}$ controls, at Day 4 to 7 post-induction; $n = 4$ technical replicates for each time-point. Data are presented as mean ± s.d. Representative data shown from one out of three independent experiments (**f, g**, and **i**). ***$P < 0.001$, ****$P < 0.0001$; two-tailed unpaired Student's t test (**f, h**, and **i**).

the expression of several well-characterized pluripotency markers at multiple time-points (up to 8 days) after $Prdm10$ deletion. Global transcriptome analysis of $Prdm10^{\Delta/\Delta}$ mESCs compared with controls at days 2 and 4 post-deletion showed no significant downregulation of genes associated with mESC pluripotency and self-renewal; in particular, the transcription factors comprising the core pluripotency regulatory circuitry ($Pou5f1$, $Klf4$, $Sox2$, $Nanog$) were expressed at levels comparable to or slightly higher relative to controls (Supplementary Fig. 4a). As further validation, we examined selected pluripotency markers ($Nanog$, $Pou5f1$, $Klf2$, $Klf4$, $Esrrb$) by qRT-PCR at day 6 and 8 post-deletion, and confirmed that their expression was maintained even at time-points where $Prdm10^{\Delta/\Delta}$ mESCs exhibit significant growth and survival defects (Supplementary Fig. 4b).

Similarly, we detected no reduction in SSEA-1 surface expression on $Prdm10$-null mESCs at day 4 and 6 post-deletion (Supplementary Fig. 4c). $Prdm10^{\Delta/\Delta}$ mESCs formed colonies smaller than that of controls, but nonetheless stained positive for alkaline phosphatase activity and showed a level of AP-positive colony formation ability comparable to that of controls, even at day 7 post-deletion (Supplementary Fig. 4d). Lastly, transcriptomic analysis of $Prdm10^{\Delta/\Delta}$ mESCs cultured under SL conditions revealed no significant misregulation of germ layer lineage markers (Supplementary Fig. 4e), confirming that loss of PRDM10 does not induce precocious differentiation. Taken together, our results indicate that PRDM10 promotes normal growth of mESCs and early embryos, but is dispensable for the maintenance of the pluripotent state.

To understand if PRDM10 may play a role in mESC differentiation, we assessed the requirement for PRDM10 during embryoid body (EB) formation (Supplementary Fig. 5a). Following LIF withdrawal, $Prdm10^{\Delta/\Delta}$ mESCs formed EBs morphologically similar to that of controls (Supplementary Fig. 5b), and upregulated expression of various germ layer lineage markers (Supplementary Fig. 5c), indicating that they retained the ability to undergo EB differentiation. Moreover, regulators of the core pluripotency network (i.e., *Pou5f1, Klf4, Nanog*) were expressed at levels comparable to or slightly higher relative to controls (Supplementary Fig. 5d), consistent to what was observed in ES cells (Supplementary Fig. 4a, b), Importantly, visible deterioration of $Prdm10^{\Delta/\Delta}$ EBs was observed at day 6 post-*Prdm10* deletion (Supplementary Fig. 5b), at a time-point similar to when $Prdm10^{\Delta/\Delta}$ mESCs cultured under pluripotency conditions also show a pronounced phenotype (Fig. 1f). Hence, our results are consistent with the essential role of PRDM10 in cell survival, and further suggest that PRDM10 is dispensable for the induction of EB differentiation.

**Genome-wide identification of PRDM10 binding sites**. Most PRDM family members can act as transcriptional regulators[2]. We thus hypothesized that the requirement for PRDM10 in mESCs may also be mediated primarily through its predicted molecular function as a sequence-specific transcription factor. To test this, we performed genome-wide profiling of PRDM10 binding sites in mESCs by chromatin immunoprecipitation with sequencing (ChIP-seq). We validated three different polyclonal antibodies against PRDM10 and used them in ChIP experiments with $Prdm10^{F/F}$; *ROSA26*-CreER$^{T2}$ mESCs to generate three independent ChIP-seq datasets, from which we identified a set of 528 reproducible peaks (IDR < 0.05) (Supplementary Fig. 6a, Supplementary Data 1). To ascertain antibody specificity, parallel experiments were performed in mESCs depleted of PRDM10 protein after 4-OHT induction. Comparison of both sets of ChIP-seq data revealed that all peaks detected in wild-type (WT) cells were absent or strongly diminished in PRDM10-depleted cells due to PRDM10 protein reduction upon recombination (Supplementary Fig. 6a). Conversely, we did not find any peaks present in PRDM10-deficient cells but absent in WT mESCs, further validating the specificity of our approach.

The PRDM10 binding sites revealed in our ChIP-seq data are strongly enriched at promoter regions, with 70.8% of peaks residing within 1 kb upstream from or overlapping with gene transcriptional start sites (TSSs) and only 9.8% mapping to intergenic regions (Fig. 2a, b). Consistent with its enrichment within gene promoters, PRDM10 binding is highly associated with regions of transcriptionally active chromatin marked by H3K4me3, H3K4me1, and H3K36me3 (Supplementary Fig. 6b).

**PRDM10 is a sequence-specific transcription factor**. By de novo motif discovery, we identified a consensus sequence highly enriched within PRDM10 binding sites (Fig. 2c). This motif showed central enrichment in PRDM10 peaks (Fig. 2d) and strong sequence conservation within PRDM10-bound sites compared with background genomic regions (Fig. 2e), leading us to hypothesize that it may be a functionally relevant candidate for DNA binding by PRDM10. To define the transcriptional impact of the sequence-specific recognition of this motif by PRDM10, we performed reporter assays in HEK293T cells transfected with constructs containing either the WT motif or a mutated version (MUT) cloned upstream of a minimal promoter to drive expression of a firefly luciferase gene (Fig. 2f). We observed strong activation of the WT motif reporter with PRDM10 over-expression; however, mutation of the consensus sequence fully

abolished PRDM10-dependent reporter activation (Fig. 2f), showing that the presence of a specific *cis*-regulatory DNA sequence is required for PRDM10-mediated transcriptional activity.

We performed gel shift assays to determine if PRDM10 binds directly to its putative DNA motif. A recombinant GST-fusion PRDM10 protein containing its central zinc finger array (GST-PRDM10$_{441-880}$) was used for binding assays. We detected robust binding of PRDM10$_{441-880}$ protein to labeled probe containing its cognate motif (Fig. 2g). This binding was specifically diminished by competition with excess unlabeled WT probe (Fig. 2h). In contrast, the MUT probe was less efficiently bound by PRDM10$_{441-880}$ in direct binding (Fig. 2g) as well as competition assays (Fig. 2h). Our findings demonstrate specific, zinc finger-mediated binding of PRDM10 to the consensus sequence identified by our ChIP-seq analysis.

**Identification of PRDM10 transcriptional activation domains**. To elucidate the mechanisms by which PRDM10 regulates transcription, we sought to characterize the function of its N-terminal PR domain and C-terminal Q-rich region. Q-rich unstructured domains have been implicated in transcriptional activation[18] and PRDM10 is the only family member harboring such domain. On the other hand, the biochemical function of the PR domain varies among PRDM family members[19], and that of PRDM10 is currently unknown. We evaluated a series of PRDM10 deletion mutants (Fig. 2i, j) for their ability to activate a luciferase reporter construct containing the PRDM10 consensus motif. In line with our gel shift assay data (Fig. 2g, h), mutants lacking the central zinc finger DNA-binding domain (ZF-DBD) failed to elicit any activity, underscoring the importance of the ZF-DBD in recruiting PRDM10 to target DNA. The ZF-DBD alone was insufficient to activate transcription, indicating a requirement for additional effector domains (Fig. 2i). Expression of the ZF-DBD in combination with either the N-terminal PR domain (N-880) or C-terminal Q-rich domain (441-C) was sufficient for transcriptional activation, albeit at lower levels compared with that achieved by full-length PRDM10 (Fig. 2i). These results demonstrate that PRDM10 engages its targets via zinc finger-mediated sequence-specific DNA binding and drives target transcription by means of N- and C-terminal activation domains.

**Gene expression changes in *Prdm10*-null mESCs and embryos**. We reasoned that the phenotypes observed in $Prdm10^{\Delta/\Delta}$ mESCs and embryos were the outcome of transcriptional misregulation following the loss of PRDM10. Therefore, we performed gene expression profiling by RNA-seq in both systems: (1) $Prdm10^{\Delta/\Delta}$ mESCs vs. vehicle-treated $Prdm10^{F/F}$ control mESCs; (2) $Prdm10^{\Delta/\Delta}$ vs. control single embryos obtained from $Prdm10^{\Delta/+}$ intercrosses. In order to predominantly capture transcriptional changes due to misregulation of primary target genes, we isolated 8-cell stage embryos at E2.5, one day before the onset of the phenotypic defects, within a developmental window where mutant embryos are morphologically indistinguishable from WT and heterozygous littermates (Supplementary Fig. 1e). For the same reason, we chose to analyze mESCs at Day 2 and 4 post-deletion (Supplementary Fig. 7a), prior to the onset of significant growth inhibition and cell death (Fig. 1f, i).

Comparing PRDM10-deficient mESCs to controls at 2 days post-deletion, we found that 8.2% ($n = 953$) of genes were significantly up- and 8.7% ($n = 1019$) were downregulated ($P_{adj} < 0.05$). More extensive transcriptional changes were observed at Day 4, with 20.7% ($n = 2393$) of genes being up- and 18.7% ($n = 2157$) being significantly downregulated ($P_{adj} < 0.05$) in PRDM10-deficient mESCs (Supplementary Fig. 7a, Supplementary Data 2

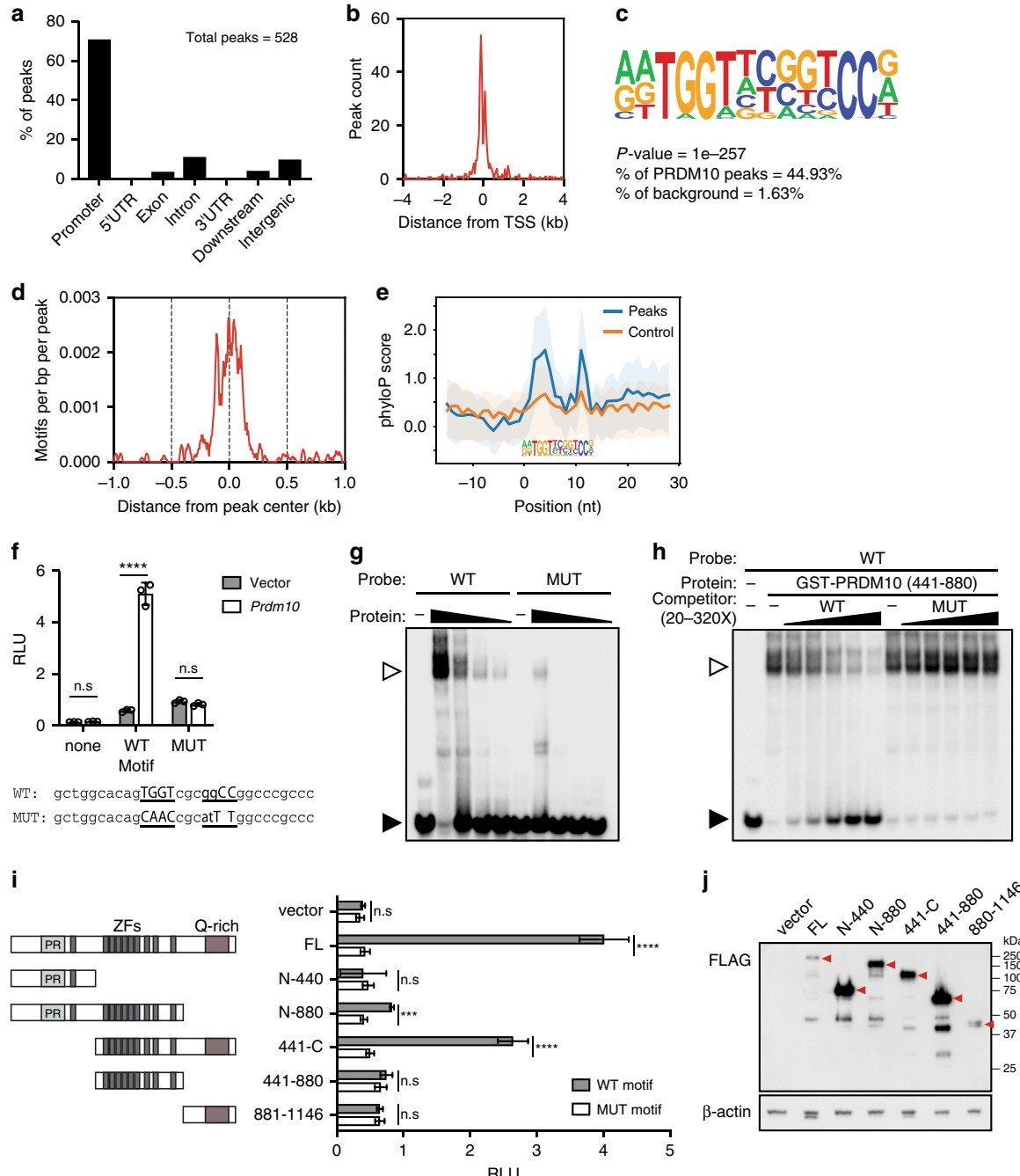

**Fig. 2 PRDM10 is a sequence-specific zinc finger transcription factor. a** Genomic feature annotation of PRDM10 binding sites ($n = 528$) reveals strong enrichment in proximal gene promoters. **b** Distribution of PRDM10 peaks relative to gene transcriptional start sites (TSS) showing that most peaks are located within 500 bp of a TSS. **c** Sequence motif identified as highly enriched in PRDM10 binding sites ($P = 10^{-257}$). **d** Density histogram showing localization of motif relative to PRDM10 peak centers. **e** Sequence conservation profiles for motifs detected within PRDM10 peaks (blue) vs. control regions (orange). Control: all genomic regions within ±1 kb of gene TSS. Y-axis: phyloP vertebrate conservation score. Shaded regions: 25–75% percentile of conservation scores. **f** PRDM10 motif validation by luciferase reporter assay. HEK293T cells were transfected with reporter constructs containing either the canonical (WT) or mutated (MUT) motif sequence, together with *Prdm10* expression plasmid or vector control. PRDM10 stimulates transcriptional activation only in the presence of the canonical motif. $n = 3$ samples. **g** Binding of labeled probe (10 nM) containing WT vs. MUT motif in the presence of PRDM10$_{441-880}$ protein (125–1000 nM), assessed by gel-shift assay. Open arrowheads: bound probe; solid arrowheads: free probe. **h** Competition assay showing specificity of PRDM10 interaction with its motif. PRDM10$_{441-880}$ binding to 10 nM labeled WT probe is diminished in the presence of 20- to 320-fold molar excess of unlabeled WT probe; competition with the MUT probe has no effect. **i** Schematic of *Prdm10* full-length and mutant expression constructs tested in reporter assays (*left*). Expression of PRDM10$_{FL}$, PRDM10$_{N-880}$, and PRDM10$_{441-C}$ selectively activates the WT motif reporter; $n = 3$ samples (*right*). **j** Western blot of FLAG-tagged PRDM10 constructs from whole cell lysates of transfected HEK293T cells. Arrowheads: proteins of interest. Loading control: β-actin. Reporter activity expressed as relative luminescence units (RLU) of firefly luminescence normalized to Renilla control (**f** and **i**). Representative data shown from one out of two (**g** and **h**) or three (**f** and **i**) independent experiments. Mean ± s.d. ***$P < 0.001$, ****$P < 0.0001$, n.s., not significant; two-tailed unpaired Student's *t* test (**f** and **i**).

and 3). Functional annotation of differentially expressed genes ($P_{adj} < 0.05$) at Day 4 post-deletion revealed a gene expression signature significantly enriched in GO terms associated with ribosomal function, translation and peptide metabolism (Supplementary Fig. 7b, Supplementary Data 4), pointing to potential misregulation of protein synthesis processes in PRDM10-deficient mESCs.

Analysis of RNA-seq data from early embryos revealed that 0.48% ($n = 50$) of genes were upregulated and 0.79% ($n = 83$) were downregulated ($P_{adj} < 0.05$) in $Prdm10^{\Delta/\Delta}$ embryos compared with $Prdm10^{+/+}$ and $Prdm10^{\Delta/+}$ controls (Supplementary Fig. 7c, Supplementary Data 5).

**PRDM10 binds and transcriptionally activates target genes**. To further narrow the list of direct and relevant candidate PRDM10 targets, we integrated ChIP- and RNA-seq data. By ChIP-seq, we identified 633 unique genes associated with PRDM10 binding sites (Supplementary Data 6). Of these, 52 and 76 were differentialy expressed ($P_{adj} < 0.05$, fold-change > 2) in mESCs at 2 and 4 days post-deletion, respectively (Fig. 3a, Supplementary Data 7). Notably, the majority of genes bound and regulated by PRDM10 ($P_{adj} < 0.05$, fold-change > 2) showed decreased expression in $Prdm10^{\Delta/\Delta}$ mESCs (Fig. 3a). In contrast, genes that were not direct targets of PRDM10 displayed expression changes in both directions and are thus likely secondary targets (Fig. 3a). The same was true for embryos, in which loss of PRDM10 led to downregulation of all 28 differentially expressed direct targets ($P_{adj} < 0.05$, fold-change > 2) (Fig. 3b, Supplementary Data 7). Taken together with our reporter assay data (Fig. 2f, i), these findings indicate that PRDM10 functions primarily as a transcriptional activator of its target genes.

**Identification of *Eif3b* as a key downstream target of PRDM10**. To determine the mechanism underlying the phenotypes observed in PRDM10-deficient mESCs and early embryos, we attempted to identify direct targets of PRDM10 that might be functionally relevant in both systems. We compared PRDM10-bound genes that were significantly downregulated ($P_{adj} < 0.05$, fold-change > 2) across all three RNA-seq datasets, and identified an overlapping set of 18 genes misregulated in both mESCs and embryos (Fig. 3c, d). Of these, seven genes are viable in the homozygous null condition, seven genes are embryonic lethal when deleted in mice, and the remaining four genes have no phenotype reported in the literature (Fig. 3d). We chose not to focus on homozygous viable genes as they were least likely to be relevant to the *Prdm10*-null embryonic lethal phenotype. Among the seven candidate genes associated with embryonic lethality, only *Eif3b*–the eukaryotic translation initiation factor 3 (eIF3) subunit B–has been implicated specifically in preimplantation development (Fig. 3d).

Homozygous deletion of *Eif3b* results in early embryonic lethality by E3.5[20], within a developmental time-frame similar to that observed in $Prdm10^{\Delta/\Delta}$ embryos. *Eif3b* encodes a highly conserved core component of the multi-subunit eIF3 complex[21], which promotes mRNA recruitment to the pre-initiation complex (PIC) as a necessary step in translation initiation[22–25]. The well-established role of *Eif3b* in translation is consistent with a gene expression signature in $Prdm10^{\Delta/\Delta}$ mESCs pointing to disrupted protein synthesis (Supplementary Fig. 7b). Furthermore, *Eif3b* has been identified as a positive hit in two genome-wide CRISPR screens for genes essential in mESCs[26] (Supplementary Fig. 8a), suggesting it is likely to act as a critical mediator of mESC survival downstream of PRDM10.

In addition to *Eif3b*, we identified two other PRDM10-regulated targets involved in translation and protein synthesis: *Eef1d* (eukaryotic translation elongation factor 1 delta) and *Rpl19* (ribosomal protein L19). *Eef1d* was strongly downregulated in *Prdm10*-null embryos as well as mESCs (Fig. 3d, Supplementary Fig. 7d, e). However, published CRISPR screens in mESCs[26] did not support an essential requirement for *Eef1d* (Supplementary Fig. 8a), and consistent with this, shRNA-mediated knockdown of *Eef1d* in mESCs had no observable effect on cell growth (Supplementary Fig. 8c, d, g). On the other hand, *Rpl19* depletion led to significantly reduced cell growth (Supplementary Fig. 8b, e, f, g), suggesting that it may also be essential for mESC viability. However, because *Rpl19* was only modestly downregulated (<2-fold) in PRDM10-deficient embryos, while *Eif3b* showed a <2-fold downregulation in vivo (Supplementary Fig. 7d), we prioritized *Eif3b* for functional validation.

**Effects of *Eif3b* depletion in embryos and mESCs**. In an effort to obtain direct evidence supporting an essential role for *Eif3b* in preimplantation development, we performed knockdown experiments both in vivo and in vitro. WT mouse zygotes were microinjected with siRNAs, and allowed to develop to blastocyst stage over 4 days in culture (Fig. 4a). Strikingly, siRNA-mediated knockdown of *Eif3b* resulted in complete developmental arrest just prior to blastocyst formation ($n = 40$): no *Eif3b*-deficient embryos were observed to form mature, cavitated blastocysts (Fig. 4a) and most exhibited a morula-like morphology (Fig. 4b). This is in agreement with previous studies indicating a pre-E3.5 lethality phenotype for *Eif3b*-null embryos[20]. In contrast, siRNA-mediated depletion of other selected PRDM10-regulated candidate genes, including *Selenow* ($n = 93$), *Ss18* ($n = 87$), and *Ube2a* ($n = 66$), showed no significant impact on embryo development (Fig. 4a).

For in vitro validation, we transfected E14 mESCs with siRNAs to knock down *Eif3b* expression (Supplementary Fig. 9a), and observed severely impaired cell growth in *Eif3b*-depleted cells compared with controls up to 72 h post-transfection (Supplementary Fig. 9b, c). Similar results were obtained utilizing a lentiviral shRNA knockdown approach, in which a 3′UTR-specific hairpin sequence (Eif3b-479) was employed to disrupt endogenous *Eif3b* expression. A modest but consistent reduction in *Eif3b* mRNA and protein levels (Fig. 4c, d) was sufficient to significantly inhibit cell growth in Eif3b-479 shRNA-transduced cells (Fig. 4e). We verified that the observed phenotype was not due to off-target effects, as it was successfully rescued by expression of shRNA-resistant *Eif3b* lacking the targeted 3′UTR (Fig. 4c–e). Collectively, these results demonstrate that *Eif3b* functions as an essential gene in early embryos as well as mESCs.

**PRDM10 directly activates transcription of *Eif3b***. Because our data showed robust PRDM10 occupancy at the *Eif3b* promoter (Fig. 5a), as well as significant downregulation of *Eif3b* transcript in $Prdm10^{\Delta/\Delta}$ mESCs and embryos (Fig. 3d, Supplementary Fig. 7d, e, Supplementary Data 3 and 5), we next investigated the hypothesis that *Eif3b* is a direct target of PRDM10-mediated regulation. From ChIP-seq data, we identified two adjacent PRDM10-binding regions (P1 and P2) within the *Eif3b* promoter (Fig. 5b), each containing a consensus motif match (Supplementary Fig. 9d). Both P1 and P2 stimulated luciferase reporter activity driven by PRDM10 overexpression (Fig. 5b), supporting their role as *cis*-regulatory sequences that recruit PRDM10 to activate *Eif3b* transcription. We further verified that $Prdm10^{\Delta/\Delta}$ mESCs had significantly reduced expression of *Eif3b* mRNA (Fig. 5c) and protein (Fig. 5d), particularly at later time-points post-deletion. Moreover, *Eif3b* transcript levels in $Prdm10^{\Delta/\Delta}$ mESCs were fully restored upon re-expression of exogenous

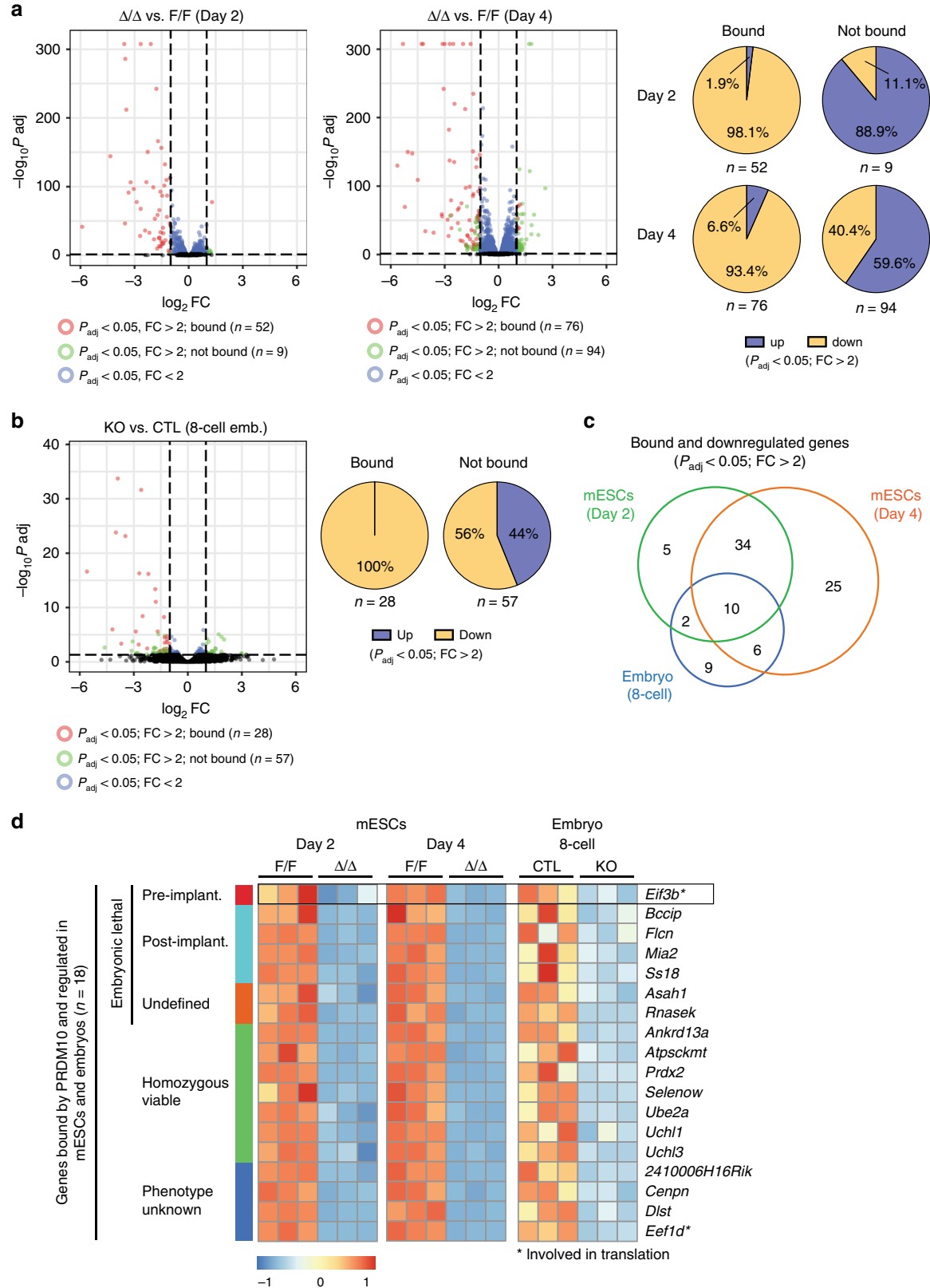

PRDM10 (Fig. 5e). Hence, PRDM10 is both necessary and sufficient to maintain normal *Eif3b* expression levels in mESCs.

**EIF3B acts downstream of PRDM10 to promote translation.** Finally, we examined the functional importance of EIF3B downstream of PRDM10, by stably overexpressing exogenous

*Eif3b* in *Prdm10*[F/F]; *ROSA26*-CreER[T2] mESCs (Fig. 6a, b), followed by 4-OHT treatment to induce *Prdm10* deletion. Strikingly, restoring *Eif3b* expression was in itself sufficient to achieve a partial phenotypic rescue in *Prdm10*[Δ/Δ] mESCs, reducing doubling time by almost 40% compared with "vector-only" control *Prdm10*-null cells (Fig. 6c, d; Supplementary Fig. 9e). The partial

**Fig. 3 PRDM10 binding is associated with transcriptional activation of target genes. a** Volcano plots of RNA-seq data from $Prdm10^{\Delta/\Delta}$ vs. $Prdm10^{F/F}$ mESCs at indicated time-points post-deletion, with $P_{adj} < 0.05$, fold-change (FC) > 2 used as the cut off to define genes with significant changes in expression. Red: genes directly bound by PRDM10; green: genes not bound by PRDM10 (*left panels*). Of these genes, PRDM10 targets are predominantly downregulated (*right*). **b** Volcano plot of RNA-seq data from $Prdm10^{\Delta/\Delta}$ (KO) vs. $Prdm10^{+/+}$ and $Prdm10^{\Delta/+}$ (CTL) 8-cell stage embryos (*left*). 28 genes bound by PRDM10 showed significant expression changes ($P_{adj} < 0.05$, fold-change > 2); of these, all were downregulated in $Prdm10$-null embryos (*right*). **c** Venn diagram depicting the overlap of bound and downregulated genes in RNA-seq datasets for embryos (8-cell) and mESCs (Day 2 and/or Day 4 post-induction), with a total of 18 genes identified as top candidates. **d** Heatmap showing all 18 bound and downregulated genes from Fig. 3c, categorized by their respective gene knockout phenotypes. Asterisks indicate genes directly involved in translation, *Eif3b* and *Eef1d*. Color scale: Z-score for row-normalized expression values, scaled separately for Day 2 p.i mESCs, Day 4 p.i mESCs, and embryos. Pre-implant: preimplantation lethality; post-implant: postimplantation lethality; undefined: developmental timing of embryonic lethality unknown.

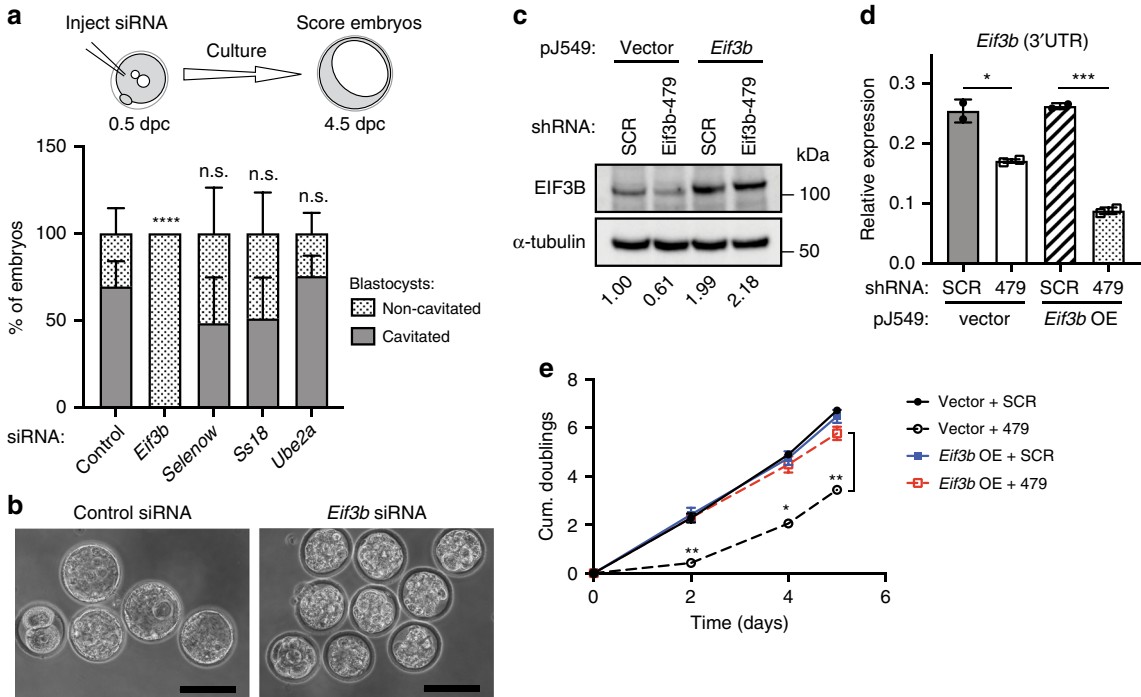

**Fig. 4 Loss of EIF3B causes lethality in preimplantation embryos and mESCs. a** *Eif3b*-deficient embryos arrest prior to blastocyst formation. Expression of candidate target genes was knocked down in wild-type zygotes by injection of siRNA: control ($n = 38$), *Eif3b* ($n = 40$), *Selenow* ($n = 93$), *Ss18* ($n = 87$), and *Ube2a* ($n = 66$), and blastocyst morphology was scored after 4 days' embryo culture. Y-axis: percentage of cavitated or non-cavitated blastocysts of total embryos analyzed in each experiment, for at least 3 independent experiments per target gene. **b** Representative images of embryos treated with control or *Eif3b*-targeting siRNA, acquired at 4.5 dpc. Scale bar: 100 μm. **c** Western blot analysis of E14 mESCs overexpressing *Eif3b* or vector control, transduced with indicated shRNA. Numbers in bottom row represent quantification of relative EIF3B protein levels after background subtraction and normalization to α-tubulin. **d** qRT-PCR validation of *Eif3b* knockdown in E14 mESCs transformed with vector control (vector) or *Eif3b* overexpression construct (*Eif3b* OE) and transduced with shRNA targeting the 3′UTR of *Eif3b* (shRNA-479). 3′UTR-specific primers were used to detect endogenous *Eif3b* transcript. $n = 2$ samples, data shown from one out of three independent experiments with similar results. **e** E14 mESCs transduced with shRNA-479 to deplete endogenous *Eif3b* exhibit slower growth (vector + 479; black dashed line), while expression of shRNA-resistant *Eif3b* restores normal growth (*Eif3b* OE + 479; red dashed line). Cells were plated at equal densities, passaged at Day 2 and Day 4, and counted at indicated time-points. $n = 3$ replicates, representative data shown (from same experiment as in Fig. 4d). Error bars denote mean ± s.d.; *$P < 0.05$, **$P < 0.01$, ***$P < 0.001$, ****$P < 0.0001$, n.s., not significant; two-tailed unpaired Student's $t$ test (**a** and **d**), two-way ANOVA with Tukey's multiple comparisons test (**e**).

nature of the phenotypic rescue by EIF3B points towards, unsurprisingly, additional contributions by other PRDM10-regulated targets (Fig. 3d, Supplementary Data 7). For example, we have identified *Rpl19* (Supplementary Fig. 8b, e) as another PRDM10-regulated target with a key role in protein synthesis that may also contribute to the growth defect in $Prdm10^{\Delta/\Delta}$ mESCs. Nonetheless, our results strongly support a critical role for EIF3B as one of the major mediators of mESC survival downstream of PRDM10, and suggest the possibility that EIF3B may also function in a similar capacity in the context of PRDM10-deficient preimplantation embryos.

Mammalian eIF3 is a large complex comprising 13 protein subunits, and is essential for stimulating multiple steps of the translation initiation pathway[22]. Given that EIF3B is one of the most highly conserved core subunits, with a critical role in the nucleation and function of the eIF3 complex[21], we reasoned that loss of EIF3B would lead to decreased global translation in $Prdm10$-null mESCs. Consistent with this hypothesis, polysome profile analysis of $Prdm10^{\Delta/\Delta}$ mESCs revealed a dramatic reduction in translation rates, with a >3-fold decrease in the polysome-to-monosome (P/M) ratio compared with $Prdm10^{F/F}$ controls (Fig. 6e). Importantly, this effect was fully rescued by re-expression of exogenous EIF3B (Fig. 6f), demonstrating that the global translation defect in $Prdm10$-null cells was specifically caused by loss of EIF3B. Though not statistically significant, we noted a trend towards slightly higher P/M ratios in

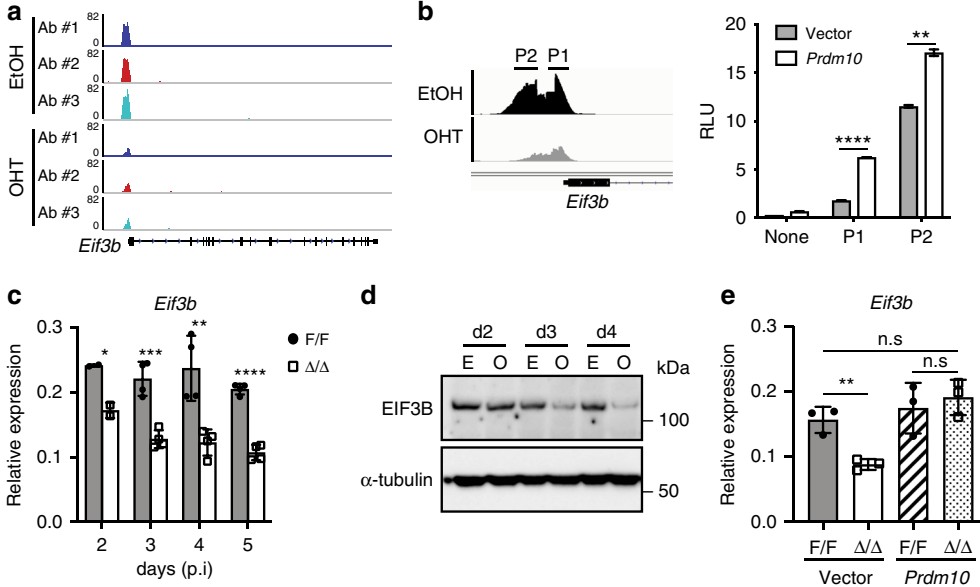

**Fig. 5 PRDM10 transcriptionally regulates *Eif3b* expression. a** ChIP-seq signal tracks showing PRDM10 occupancy at the *Eif3b* promoter in EtOH- vs. OHT-treated *Prdm10*[F/F]; CreER[T2] mESCs, analyzed at 2 days post-induction. Vertical axis: fold-change over input. **b** PRDM10 enrichment at the *Eif3b* promoter is detected as two subpeaks, denoted P1 and P2 (*left*). PRDM10-dependent activation of luciferase reporters containing *Eif3b* promoter sequences derived from subpeak P1 or P2 (*right*); *n* = 2 samples. Representative data shown from one out of three independent experiments. **c** qRT-PCR quantification of changes in *Eif3b* transcript at indicated time-points after *Prdm10* deletion; *n* = 2–3 biological replicates. **d** Western blot analysis of EIF3B protein levels in *Prdm10*[F/F]; CreER[T2] mESCs at indicated time-points after treatment with EtOH (E) or 4-OHT (O). **e** qRT-PCR analysis showing restoration of *Eif3b* transcript levels in *Prdm10*[Δ/Δ] mESCs by exogenous *Prdm10*. Data shown at Day 4 post-induction; *n* = 3. Gene expression shown relative to *Ubb* (**c** and **e**). Error bars denote mean ± s.d.; *$P < 0.05$, **$P < 0.01$, ***$P < 0.001$, ****$P < 0.0001$, n.s., not significant; two-tailed unpaired Student's *t* test (**b**, **c** and **e**).

EIF3B-overexpressing cells compared with vector-transduced *Prdm10*[F/F] controls, suggestive of translation rates above baseline and consistent with previous reports that overexpression of the EIF3B subunit is sufficient to elevate levels of the entire eIF3 complex, thereby activating protein synthesis in cancer cell lines[27]. Taken together, our data strongly suggest that the growth defect observed in *Prdm10*[Δ/Δ] mESCs is a functional consequence of decreased translation efficiency arising from misregulation of *Eif3b*.

## Discussion

In this study, we demonstrate an essential role for PRDM10 as a transcriptional regulator in early mammalian embryogenesis and mESC homeostasis. Our findings strongly support a model whereby PRDM10 supports cell growth and survival during early development by transcriptionally regulating *Eif3b* expression to sustain global translation. Although this work focuses on phenotypes related to early development, this does not imply that the function of PRDM10 is strictly specific to early embryogenesis, or that PRDM10 regulates processes unique to mESCs. Given that PRDM10 is expressed across multiple tissues and regulates a broad range of target genes, it is highly likely to have pleiotropic effects that may be revealed using different models for conditional deletion at later developmental stages or in specific tissues. Interestingly, PRDM10 gene fusions have been implicated in the pathogenesis of undifferentiated pleomorphic sarcoma[15,16], raising the possibility of other yet-unknown roles in human disease. Hence, the data presented in this work provide a strong starting point for future studies to further extend our understanding of PRDM10 in development and disease.

The PRDM family first appeared in metazoans, and *Prdm10* is thought to have evolved during a family expansion just prior to vertebrate evolution. PRDM genes that emerged later in evolution tend to be specifically expressed in highly specialized cells to serve tissue-specific functions; examples include *Prdm14* (germ cells and embryonic stem cells), *Prdm7* (melanocytes), and *Prdm9* (testis)[28]. Surprisingly, despite being one of fastest evolving paralogs, *Prdm10* is expressed across a broad range of adult tissues[28], and is the first example of a PRDM family member being implicated in early embryo development prior to mid-gestation. It is also interesting to note that while *Prdm10* evolved fairly recently, the eIF3 complex arose before metazoan evolution, and EIF3B is one of five eIF3 subunits that are conserved in all eukaryotes. The significance and implications of PRDM10 regulating such an evolutionarily ancient gene remains an open question. Evolutionary expansion and divergence within zinc finger protein families drives diversification of DNA binding specificities and effector functions, and is highly correlated with increasing organism complexity during vertebrate evolution[29]. Recent findings have challenged the traditional view of eIF3 as a general translation initiation factor, suggesting that eIF3 may also exert selective translational control over specific mRNAs[30]. It is therefore intriguing to speculate that PRDM10 may contribute additional layers of regulatory control over *Eif3b* expression in a context-dependent or tissue-specific manner, or during different developmental stages.

Although protein synthesis has traditionally been regarded as a basic cellular housekeeping function, emerging evidence suggests that it is dynamically regulated during development and plays a significant role in the maintenance of stem cells[31–34]. Furthermore, it has been reported that EIF3B, along with other proteins involved in translation initiation, is upregulated at the morula and blastocyst stages, suggesting a key role for protein synthesis in supporting embryonic growth during this developmental transition[35]. Notably, multiple studies have demonstrated the importance of global translational control in ESC pluripotency and differentiation[36–40]. However, despite major advances in our understanding of the transcriptional and epigenetic mechanisms governing ESC function[41,42], much less is known about how

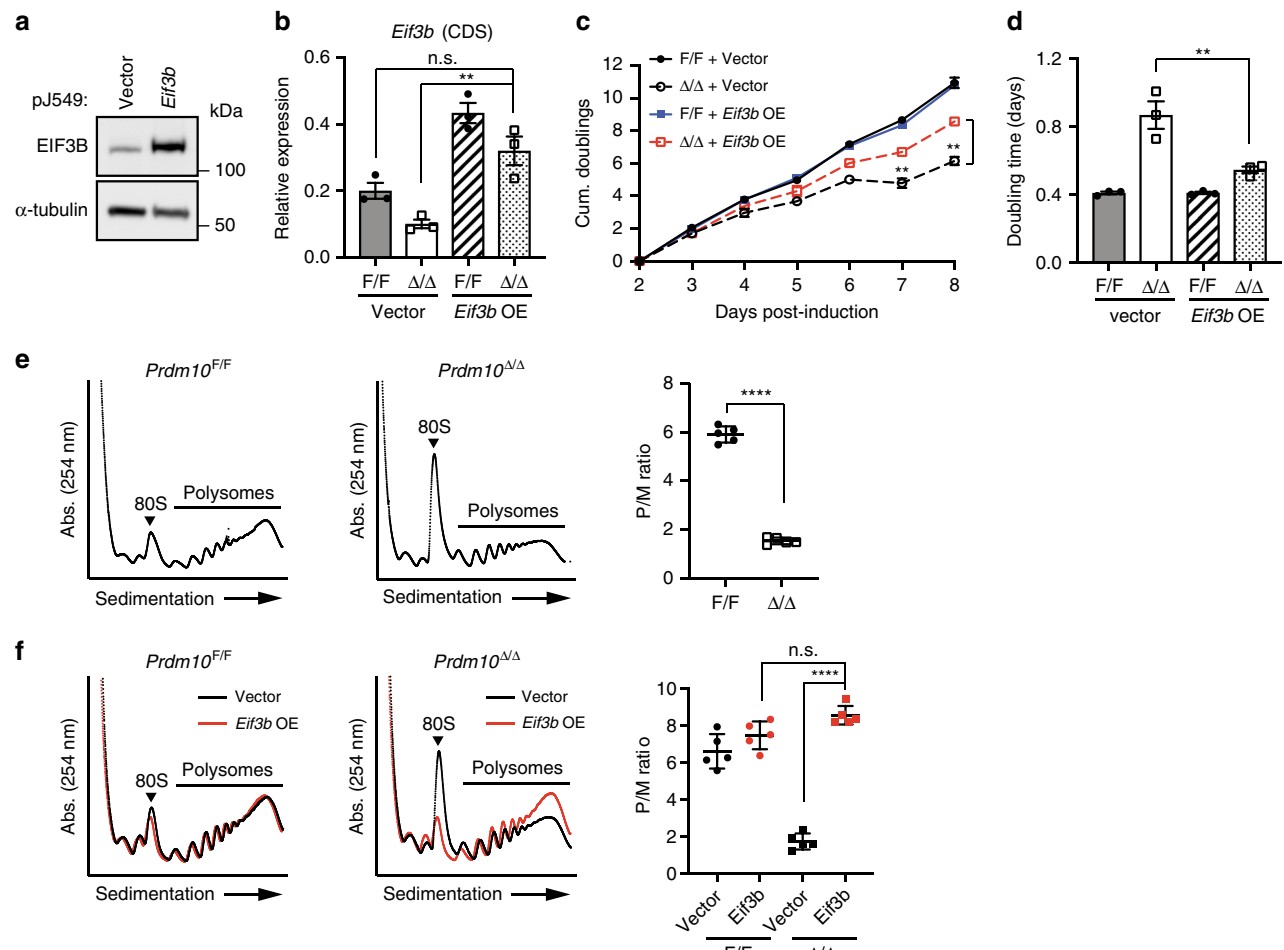

**Fig. 6 EIF3B promotes global translation downstream of PRDM10. a** Immunoblotting detection of FLAG-tagged EIF3B protein expression in *Prdm10*F/F; CreER[T2] mESCs transformed with pJ549-*Eif3b* or vector. **b** qRT-PCR validation of *Eif3b* transcript in *Prdm10*F/F; CreER[T2] mESCs transformed with pJ549-*Eif3b* or vector, analyzed 6 days post-induction. CDS-specific primers were used to detect both endogenous and overexpressed *Eif3b* transcript; $n = 3$ biological replicates. **c** Growth curve analysis of *Prdm10*F/F and *Prdm10*Δ/Δ mESCs in the presence of empty vector or exogenous *Eif3b*, showing partial phenotypic rescue by *Eif3b* overexpression. Cells were passaged at constant density every 2 days and counted daily up to Day 8 post-induction; $n = 3$. Representative data shown from one of three independent experiments. **d** Average population doubling times over the course of the growth assay for *Prdm10*F/F and *Prdm10*Δ/Δ mESCs expressing vector control or exogenous *Eif3b*. Each point represents data from one independent experiment; $n = 3$. **e** Representative polysome profiles of *Prdm10*F/F and *Prdm10*Δ/Δ mESCs analyzed 5 days post-induction (*left*). A significantly reduced polysome:monosome (P/M) ratio indicates impaired global translation in *Prdm10*Δ/Δ mESCs (*right*); $n = 5$ biological replicates, across two independent experiments. **f** Representative polysome profiles of *Prdm10*F/F and *Prdm10*Δ/Δ mESCs expressing *Eif3b* or vector, analyzed at Day 5 post-induction (*left*). *Eif3b* overexpression restores global translation in *Prdm10*Δ/Δ mESCs to levels comparable to *Prdm10*F/F controls, as quantified by P/M ratio (*right*); $n = 5$ biological replicates across two independent experiments. Gene expression presented relative to *Ubb* (**b**). Data presented as mean ± s.d.; **$P < 0.01$, ****$P < 0.0001$, n.s., not significant; one-way ANOVA with Tukey's multiple comparisons test (**b**, **d** and **f**); two-way ANOVA with Tukey's multiple comparisons test (**c**).

regulation is achieved at the level of protein translation. As translation is mostly regulated at the initiation stage, eukaryotic initiation factors have been extensively characterized in terms of their biochemical and structural properties[23]; however, few studies have examined their potential roles in stem cells and development. By deciphering the molecular mechanisms underlying the requirement for PRDM10 in mESCs, our work provides insight into how precise transcriptional control of the translation machinery may modulate global translation to influence development.

## Methods

**Mice**. *Prdm10* conditional knockout mice were generated on a C57BL/6 background using an ES cell clone (Prdm10[tm1a(EUCOMM)Hmgu]) from the EUCOMM consortium containing a knockout-first cassette targeting exon 5 of *Prdm10*. After successful germline transmission, *Prdm10*[lacZ/+] mice were crossed to a FLPe

recombinase transgenic line (C57BL/6-Tg(CAG-Flpe)2Arte, Taconic) to remove the neomycin selection cassette and generate the Prdm10 flox allele. *Prdm10*F/+ mice were then bred to an ACTB-Cre recombinase transgenic line to generate the *Prdm10*Δ null allele. 4-OHT-inducible knockouts were created by crossing *Prdm10*F/F mice with a *ROSA26*-CreER[T2] transgenic strain. Mice were housed in specific pathogen-free conditions and maintained on a 12 h light-dark cycle with food and water available *ad libitum*. All procedures involving mice were performed in compliance with the Institutional Animal Care and Use Committee protocols #151042 and #181393, with the approval of the Biological Resource Centre (BRC), A*STAR.

**Genotyping**. For genotyping of mice, crude DNA extracts were prepared from tail biopsies by overnight lysis at 55 °C in DirectPCR Lysis Reagent (Viagen) containing 0.2 mg/ml proteinase K. For genotyping of embryos, each embryo was lysed for 1 h at 55 °C in 10 µl lysis buffer (50 mM KCl, 10 mM Tris-HCl pH 8.3, 2 mM MgCl2, 0.45% NP-40, 0.45% Tween-20, 0.5 mg/ml proteinase K), followed by heat-inactivation at 95 °C for 10 min. 2 µl extract was used as template per 20 µl PCR

reaction with 1X DreamTaq PCR master mix (Thermo Fisher Scientific). Primers used for genotyping are listed in Supplementary Data 8.

**Plasmid construction.** For overexpression studies, full-length *Prdm10* was cloned by gene synthesis (AITBiotech) based on sequence information obtained from RNA-seq analysis and Ensembl transcript annotations. For stable expression in mESCs, *Prdm10* was ligated into the NheI and BamHI sites of the pJ549 PiggyBac transposase expression vector (DNA 2.0), modified to contain an N-terminal FLAG tag. For transient transfections, *Prdm10* was cloned into pcDNA3.1 (Invitrogen) downstream of an N-terminal FLAG tag via EcoRI and NotI restriction sites. For the construction of shRNA lentiviral vectors, gene-specific hairpin sequences were selected from the RNAi Consortium TRC lentiviral shRNA library and cloned into pLKO.1-Neo (Addgene, #13425) as annealed oligonucleotides (Supplementary Data 8). Endotoxin-free plasmids were prepared using the Nucleobond Xtra Midi EF Kit (Macherey-Nagel). All constructs were verified by Sanger sequencing.

**Embryo isolation and culture.** Natural matings were set up and successful copulation (assumed to have occurred at 12 midnight) was ascertained by the presence of a vaginal plug. Pregnant females were sacrificed by cervical dislocation at 36 h post-copulation (E1.5) for 2-cell embryos and 84–90 h post-copulation (E3.5) for blastocysts. Preimplantation embryos were collected from the infundibulum (E1.5) or uterine horns (E3.5) by flushing with M2 medium (Millipore). Postimplantation embryos (E4.5–12.5) were dissected from the uteri. For in vitro studies, embryos were isolated at the 2-cell stage, cultured for 3 days in KSOM + AA medium (Millipore) at 37 °C with 5% CO$_2$, and imaged at indicated time-points. Data were pooled from heterozygous intercrosses with *Prdm10*$^{lacZ/+}$ and *Prdm10*$^{Δ/+}$ mice as both mutant alleles yielded equivalent phenotypic outcomes.

**Immunofluorescence (IF) microscopy.** Embryos were fixed in 4% paraformaldehyde for 30 min and permeabilized with 0.1% Triton X-100 for 30 min at room temperature. After blocking with 1% FBS, samples were incubated at 4 °C overnight using the following primary antibodies and dilutions: OCT4 (1:100; sc-5279, Santa Cruz), CDX2 (1:100; ab88129, Abcam), and NANOG (1:100; RCAB002P-F, ReproCELL). Samples were then washed and incubated in Alexa Fluor-conjugated secondary antibodies (1:500; Life Technologies) at room temperature for 1 h. Images were acquired on either an Olympus Fluorview 1000 (60× oil immersion objective) or Zeiss LSM800 (63× oil immersion objective) confocal laser-scanning microscope.

**Derivation of mouse ES cell lines.** Blastocysts were isolated at E3.5 and cultured in 2i/LIF medium on a feeder layer of mitotically inactivated mouse embryonic fibroblasts (MEFs). After 5 days, ICM outgrowths were disaggregated using a fine Pasteur pipette with trypsin-EDTA treatment, re-plated on MEF feeders in 2i/LIF medium, and gradually expanded over 4–5 passages. 2i/LIF medium comprised a 1:1 mix of DMEM/F12 (Gibco) and Neurobasal (Gibco) media, supplemented with GlutaMAX (Gibco), 0.05% bovine serum albumin (BSA), 100 U/ml penicillin/streptomycin, N2 (Gibco), B27 (Gibco), 10 μM 2-mercaptoethanol (Gibco), 1000 U/ml LIF (Millipore), 3 mM CHIR99021 (Axon), and 1 mM PD0325901 (Axon). To determine genotypes of newly-derived mESC lines, cells were plated on 0.1% gelatin for 1 h to allow feeder MEFs to preferentially attach, after which MEF-depleted ESCs were collected and genomic DNA purified using the DNeasy Blood & Tissue kit (Qiagen).

**Cell culture and transfection.** All cell cultures were maintained at 37 °C in a humidified incubator with 5% CO$_2$. mESCs were cultured on 0.1% gelatin-coated plates in Dulbecco's Modified Eagle's Medium (DMEM, Hyclone) supplemented with 15% fetal bovine serum (FBS), GlutaMAX, nonessential amino acids, 1 mM sodium pyruvate, 100 U/ml penicillin/streptomycin, 5.5 μM 2-mercaptoethanol and 1000 U/ml mLIF (Millipore). For routine propagation, cells were trypsinized, resuspended in 10% FBS DMEM, and re-seeded at a ratio of 1:20–1:40 every 2–3 days. Plasmid and siRNA transfections into mESCs and HEK293T cells were performed using Lipofectamine 2000 (Thermo Fisher Scientific). All cell lines used in this study tested negative for mycoplasma contamination by a PCR-based assay (Biological Industries).

**Induction of Prdm10 deletion.** *Prdm10*$^{F/F}$; *ROSA26*-CreER$^{T2}$ mESCs were treated for 24 h with 50 nM 4-OHT (Sigma) or an equal volume of ethanol. Cells were then washed in dPBS and seeded for assays.

**Generation of cell lines with stable transgene expression.** Mouse ESCs were transfected with pJ549 PiggyBac expression plasmids (DNA 2.0) by reverse transfection with Lipofectamine 2000. Transfected E14 mESCs were selected in 0.75 μg/ml puromycin for 3 days. For stable transgenesis of *Prdm10*$^{F/F}$; *ROSA26*-CreER$^{T2}$ mESCs, GFP-positive cells were purified by fluorescence-activated cell sorting on a MoFlo XDP cell sorter (Beckman Coulter) and expanded in culture for assays.

**Apoptosis assay.** One day before the indicated time-points, mESCs were seeded in a 96-well plate at a density of $2 \times 10^4$ cells per well in quadruplicate, and allowed to attach overnight. To detect caspase activity, an equal volume of Caspase-Glo 3/7 Assay reagent (Promega) was added to each well and cells were homogenized by gentle agitation. Samples were transferred to a 96-well white flat bottom plate (Corning), incubated for 30 min at room temperature, and luminescence readings acquired on a GloMax instrument (Promega). For each condition, background-subtracted Caspase-Glo signals in each well were normalized to cell numbers, which were measured in parallel using the CellTiter-Glo Cell Viability Assay (Promega).

**Cell growth analysis.** mESCs were seeded in triplicate at a density of $1 \times 10^5$ cells per well in a 12-well plate. At indicated time-points, cells were harvested by trypsinization and total viable cell counts were measured by trypan blue exclusion on the Countess II Automated Cell Counter (Thermo Fisher). Brightfield images were acquired on a Nikon Eclipse TS100 inverted microscope with a 4× objective.

**Colony formation assay.** mESCs were seeded in 6-well plates at a density of $1 \times 10^4$/well and cultured for 5 days with regular medium changes. Cells were then washed twice in dPBS, fixed in ice-cold methanol for 10 min and stained with 0.5% crystal violet solution in 25% methanol. After washing and air-drying, colonies were imaged on a flatbed scanner and quantified using Fiji/ ImageJ.

**shRNA lentiviral production and transduction.** For lentiviral production, every 1 μg of pLKO-Neo plasmid was co-transfected with 0.5 μg pMD2.G (Addgene) and 0.375 μg psPAX2 (Addgene) into HEK293T cells using Lipofectamine 2000. Viral supernatants were collected 48 and 72 h post-transfection, filtered through a 0.45 μm syringe filter, and centrifuged at 24,000 r.p.m. for 2 h at 4 °C. The viral pellet was reconstituted in Hank's Balanced Salt Solution and stored in aliquots at −80 °C. E14 mESCs were infected with lentivirus supplemented with 8 μg/ml polybrene (Merck Millipore) and cultured in medium containing 400 μg/ml G418 (Invivogen) to select for stable transductants. After 96 h of selection, cells were re-seeded and maintained in 200 μg/ml G418 for the duration of the assays.

**siRNA-mediated knockdown in mESCs.** $2.5 \times 10^4$ E14 mESCs were transfected in suspension with 20 pmol ON-TARGETplus SMARTpool siRNA (Dharmacon) and 1 μl Lipofectamine 2000 and seeded in 24-well plates. Cell viability was assessed at 48 and 72 h post-transfection.

**RNA extraction and qRT-PCR.** Samples were lysed in TRIzol (Invitrogen) and total RNA was purified using the PureLink RNA Mini Kit (Invitrogen) with on-column DNase treatment according to manufacturer's instructions. Reverse transcription was performed using the Maxima First Strand cDNA Synthesis Kit (Thermo Scientific) with ~1 μg RNA per reaction. cDNA was diluted 10-fold with nuclease-free water for use in downstream assays. Quantitative real-time PCR was performed on a CFX96 Touch Real-Time PCR Detection System (Bio-Rad) using PowerUp SYBR Green Master Mix (Applied Biosystems). Target gene expression relative to an internal reference gene (*Ubb*) was calculated using $2^{-ΔCt}$. Primer sets were validated for specificity by melt-curve analysis and tested for linear amplification over four orders of magnitude. Primers used for qRT-PCR are listed in Supplementary Data 8.

**Western blot analysis.** Whole cell lysates were prepared in reducing sample buffer (32.9 mM Tris-HCl, 12.5% glycerol, 1% SDS, 2.5% 2-mercaptoethanol, 27 mg/ml DTT, 0.005% bromophenol blue) and heated at 98 °C for 10 min. Protein concentrations were measured using the RC DC Protein Assay (Bio-Rad). 20–40 μg total protein was loaded per well and samples were separated by SDS-PAGE gel electrophoresis in Tris-Glycine-SDS buffer (1st Base). Proteins were then transferred to Immun-Blot PVDF membranes (Bio-Rad) by wet electroblotting in Tris-glycine buffer containing 10% methanol. Membranes were blocked for 1 h at room temperature in TBS-T (Tris-buffered saline + 0.05% Tween-20) containing 5% milk or 3% BSA. Blots were incubated overnight at 4 °C with the following primary antibodies diluted in 3% BSA/TBS-T:PRDM10 (1:1000; A303-204A, Bethyl Laboratories), EIF3B (1:1000; VPA00380, Bio-Rad), FLAG M2 (1:1000, F1804, Sigma), alpha-tubulin (1:10,000; T5168, Sigma), beta-actin (1:1000; sc-47778, Santa Cruz). After three washes in TBS-T, blots were incubated with HRP-conjugated anti-mouse (1:10,000; sc-516102, Santa Cruz) or anti-rabbit (1:10,000; sc-2357, Santa Cruz) secondary antibody for 1 h at room temperature. SuperSignal West Femto Maximum Sensitivity Substrate or SuperSignal West Pico Chemiluminescent Substrate (Thermo Fisher Scientific) was used for chemiluminescent detection. Blots were imaged on a ChemiDoc Touch (Bio-Rad) and analyzed with Image Lab software (Bio-Rad).

**Luciferase reporter assays.** To characterize the PRDM10 binding motif, oligonucleotides containing the motif sequence were annealed and ligated into the NheI and BglII sites of pGL4.23 [luc2/minP] (Promega). *Eif3b* promoter sequences were amplified by PCR from mouse genomic DNA and cloned into pGL4.23 using KpnI and XhoI. pGL4.23 reporter constructs were co-transfected with pGL4.74 [hRluc/

TK] and pcDNA3-*Prdm10* expression plasmids into HEK293T cells cultured on 12-well plates. At 48 h post-transfection, firefly and Renilla luciferase activities were measured on a GloMax luminometer (Promega) using the Dual-Luciferase Reporter Assay System (Promega).

**EMSA.** Labeled probes (Supplementary Data 8) were prepared by annealing 5′-biotinylated oligonucleotides (IDT). Binding reactions with purified recombinant protein and 10 nM labeled probe were performed at room temperature for 20 min in buffer containing 6 mM HEPES-KOH pH 7.9, 6% (v/v) glycerol, 100 µg/mL BSA, 0.4 µM $ZnCl_2$ and 20 µg/mL poly(dI-dC). Samples were electrophoresed on 6% native polyacrylamide gels in 0.25× TBE buffer at 150 V, and transferred to Hybond N + nylon membranes (Amersham Pharmacia Biotech) in 0.5× TBE at 380 mA for 1 h. Biotinylated DNA was detected using the LightShift Chemiluminescent EMSA kit (Thermo Fisher Scientific). Blots were imaged on a ChemiDoc Touch (Bio-Rad) and analyzed using Image Lab (Bio-Rad).

**Flow cytometry.** For cell cycle analysis, mESCs were seeded in six-well plates such that they were 40–60% confluent on the day of harvest. Cells were pulsed with 10 µM 5-ethynyl-2′-deoxyuridine (EdU) for 30 min, washed with PBS and dissociated into a single cell suspension by trypsin-EDTA. Fixation, permeabilization and EdU labeling was performed using the Click-iT Plus EdU Alexa Fluor 488 Flow Cytometry Assay Kit (Thermo Fisher Scientific) according to manufacturer's instructions. Cells were incubated in 10 µg/ml DAPI on ice for >1 h to stain DNA and filtered through a 40 µm cell strainer immediately before data acquisition. To assess SSEA-1 surface expression, staining was carried out in FACS buffer (2% FBS in PBS) using mouse anti-SSEA-1 conjugated to Alexa Fluor 647 (BD Pharmingen, clone MC480) with 30 min incubation on ice. All flow cytometry data were acquired on a BD LSR II flow cytometer (BD Biosciences) and analyzed using FlowJo v10 (FlowJo LLC).

**Polysome fractionation.** At Day 3 post-induction, mESCs were seeded at a density of $1 × 10^6$ (*Prdm10*^F/F^) or $2.5 × 10^6$ (*Prdm10*^Δ/Δ^) cells per 10-cm dish such that they were at similar confluence for analysis at Day 5. Cells were treated with 100 µg/ml cycloheximide (Sigma) for 10 min at 37 °C to arrest translation, then washed in ice-cold PBS and harvested on ice by scraping in 800 µl lysis buffer (10 mM Tris-HCl pH 7.4, 5 mM $MgCl_2$, 100 mM KCl, 1% Triton X-100) supplemented with 2 mM DTT, 100 U/ml RNasin (Promega), protease inhibitor cocktail and 100 µg/ml cycloheximide. Lysates were sheared using a 26G needle and cleared by centrifuging at $1300 × g$ for 10 min at 4 °C. Clarified lysates were layered onto 10–50% sucrose gradients and centrifuged in an SW-41Ti rotor at 36,000 r.p.m. for 2 h. Gradients were fractionated using a BioComp Gradient Station fractionator, and absorbance at 254 nm was monitored to obtain the polysome profile. Polysome/monosome (P/M) ratios were derived by integrating the area under the respective peaks using Microsoft Excel.

**ChIP-seq.** Feeder-free mESCs were harvested by trypsinization at 48 h post-induction and resuspended to $5 × 10^6$ cells/ml in 10% FBS DMEM. Cells were cross-linked with 1% formaldehyde for 15 min at room temperature, quenched with 125 mM glycine and washed twice in cold PBS. Chromatin extracts were obtained by successive rounds of lysis in LB1 (50 mM HEPES-KOH pH 7.5, 140 mM NaCl, 1 mM EDTA, 10% glycerol, 0.5% Nonidet-P40, 0.25% Triton X-100), LB2 (10 mM Tris-HCl pH 8.0, 200 mM NaCl, 1 mM EDTA, 0.5 mM EGTA) and LB3 (10 mM Tris-HCl pH 8.0, 100 mM NaCl, 1 mM EDTA, 0.5 mM EGTA, 0.1% sodium deoxycholate, 0.5% N-lauroylsarcosine), supplemented with 0.2 mM PMSF and protease inhibitor cocktail. Chromatin DNA was sheared to a size range of 100–500 bp with 5–6 cycles of sonication at 30% amplitude using a Branson Digital Sonifier (S540D). Triton X-100 was added to a final concentration of 1% and lysates were cleared by centrifugation. 5 µg antibody was added to 100 µg of sonicated chromatin and incubated overnight with rotation at 4 °C. 40 µl Protein A Dynabeads were added to each reaction and incubated at 4 °C for 4 h. Beads were then collected on a magnetic rack and washed in low salt buffer (10 mM Tris-HCl pH 8.0, 150 mM NaCl, 1 mM EDTA, 1% Triton X-100), high salt buffer (20 mM Tris-HCl pH 8.0, 500 mM NaCl, 2 mM EDTA, 1% Triton X-100, 0.1% SDS), LiCl buffer (10 mM Tris-HCl pH 8.0, 250 mM LiCl, 1 mM EDTA, 0.5% sodium deoxycholate, 0.5% Nonidet-P40), and TE buffer with 50 mM NaCl, then incubated with elution buffer (50 mM Tris-HCl pH 8.0, 10 mM EDTA, 1% SDS) at 65 °C for 20 min with continuous agitation. Eluted protein/DNA complexes were reverse-crosslinked overnight at 65 °C, treated with RNase A (Sigma) and proteinase K, and ChIP DNA was column-purified using QIAquick PCR Purification Kit (Qiagen). DNA concentrations of input and immunoprecipitated samples were measured on a Qubit instrument with Qubit dsDNA HS Assay kit (Thermo Fisher Scientific). Libraries were prepared from 4 ng of ChIP DNA using the NEBNext Ultra II DNA Library Preparation Kit for Illumina (NEB), largely following manufacturer's instructions. All libraries were amplified for 9 PCR cycles and final elution volumes were reduced. Libraries were quantified by High Sensitivity DNA Assay on a Bioanalyzer and quantitative real-time PCR (KAPA Library Quantification Kit for Illumina, Roche). Final ChIP-Seq libraries were pooled and sequenced on an Illumina NextSeq 500 using single-end 75 bp reads to generate ~20 M raw reads per library.

**ChIP-seq data analysis.** ChIP-seq data were processed using the ENCODE Transcription Factor and Histone ChIP-Seq pipeline (https://github.com/ENCODE-DCC/chip-seq-pipeline2; v1.1.7, commit 2f567e6e). Briefly, raw fastq files were aligned with bwa (v0.7.13) then filtered to remove duplicates (with Picard v2.10.6), multi-mapping reads and low-quality alignments (with samtools v1.2). SPP (v1.13) was then used to call peaks from the filtered alignments; in addition to calling peaks on each individual sample, peaks were called on the pooled set of alignments across all samples, as well as two pseudo-replicates obtained from the pooled set by splitting alignments into two equal sets. Peaks in the ENCODE mm10 blacklist were filtered out. The filtered peak sets were then assessed for reproducibility using IDR[43] (v2.0.4.2). The final peak set used was the "optimal" peak set from the pipeline, which was obtained from the pooled pseudo-replicates with IDR cut-off of 0.05. ChIP-seq peak annotation analyses were carried out using the R package ChIPpeakAnno[44] (v3.16.1). Using the *annoPeaks* function with GRCm38.p6 annotations, peaks were annotated with genes if they were located within 5 kb upstream to 1 kb downstream of the gene body. Distribution of peaks over genomic features were summarized in peak-centric view using the function *assignChromosomeRegion*. ChIP-seq signal tracks were visualized using the Integrative Genomics Viewer[45] (IGV, v2.4.14) and heatmaps were generated using SeqPlots[46]. Histone ChIP-seq datasets with the identifiers ENCFF043LTY (H3K4me1), ENCFF469DBC (H3K4me3), ENCFF289ATH (H3K9me3), ENCFF012GHA (H3K27me3), and ENCFF785WPG (H3K36me3) were downloaded from the ENCODE portal[47].

**Motif analysis.** Homer[48] (v4.10.4; findMotifsGenome.pl) was used to discover motifs from PRDM10 ChIP-seq peaks. Repeat-masked sequence from the mm10 assembly (mm10r) was extracted from peak regions (−size given) defined in the optimal peak set (pooled pseudo-replicates, IDR cutoff 0.05, blacklist filtered) and used to discover up to ten motifs (−S 10) for lengths from 8 to 20 bp with a step of 2 (−len 8,10,12,14,16,18,20), allowing up to four mismatches in the optimization (−mis 4). Homer (annotatePeaks.pl) was then used to scan the ChIP-seq peaks (−size given) as well as TSS ± 1 kb regions (tss mode, −size −1000,1000) for the discovered PRDM10 motif to obtain a set of hits for the conservation analysis. deeptools computeMatrix (v3.3.0) was used to extract phyloP scores 15 bp upstream and downstream of the motif hit sites; phyloP scores are for multiple alignments with 59 vertebrate genomes to the mouse genome and were obtained from the UCSC genome browser (http://hgdownload.cse.ucsc.edu/goldenpath/mm10/phyloP60way/mm10.60way.phyloP60way.bw). The scores were subsequently plotted with the Python package seaborn.

**mESC RNA-seq.** *Prdm10*^F/F^; *ROSA26*-CreER^T2^ mESCs were harvested in TRIzol at indicated time-points post-induction and total RNA purified as described above. Total RNA was quantified on a Nanodrop and RNA quality was assessed on an Agilent 2100 Bioanalyzer using the RNA 6000 Nano kit (Agilent Technologies). RNA integrity values were between RIN9.5–10, confirming high quality total RNA. RNA-Seq libraries were constructed from 1 mg total RNA using the TruSeq Stranded Total RNA Sample Preparation kit (Illumina) following manufacturer's instructions with a few modifications. The resulting libraries were assessed by High Sensitivity DNA Assay on a Bioanalyzer and quantitative real-time PCR (KAPA Library Quantification Kit for Illumina, Roche). Final RNA-Seq libraries were pooled and sequenced on an Illumina NextSeq 500 instrument using paired-end $2 × 75$ bp reads to generate >48 M raw reads per sample.

**Single embryo RNA-seq.** Naturally mated timed pregnant females from *Prdm10*^Δ/+^ intercrosses were sacrificed at E2.5. 8-cell stage single embryos were collected and individually snap frozen in 4 ul lysis buffer containing dNTP mix, oligo-dT primer and SUPERase-In RNase inhibitor (Invitrogen). RNA-seq library preparation was performed according to the Smart-seq2 protocol with slight modifications[49]. Embryos were isolated and processed in two separate batches. To pre-screen cDNA samples for downstream library preparation, mutant (*Prdm10*^Δ/Δ^) and control (*Prdm10*^+/+^, *Prdm10*^Δ/+^) embryos were identified by qRT-PCR detection of *Prdm10* exon 5. 16 cycles of PCR pre-amplification were carried out for batch 1 and 13 cycles for batch 2. For both batches, 1 ng of cDNA was used for tagmentation with the Nextera XT DNA Library Prep Kit (Illumina) as described[49]. Final libraries were amplified and pooled for single-end sequencing on an Illumina NextSeq 500 instrument with 75 bp read length to yield ~4.5–10.6 M raw reads per embryo.

**RNA-seq data analysis.** RNA-seq data was processed using the ENCODE STAR-RSEM pipeline (https://github.com/ENCODE-DCC/long-rna-seq-pipeline/blob/master/DAC/STAR_RSEM.sh). In brief, sequence reads were mapped to the mouse genome build GRCm38.p6 using STAR[50] (v2.6.0c), and gene-level transcript abundances were quantified by RSEM[51] (v1.3.1) against the GENCODE mouse vM18 annotation set. DESeq2[52] (v1.22.2) was used for differential expression analysis; genes were considered to be significantly differentially expressed at $P_{adj} < 0.05$, and a minimum expression threshold was applied to exclude low-abundance genes (mESCs: baseMean > 100; embryos: baseMean > 10). Volcano plots were generated using the R package EnhancedVolcano (v.1.0.1), expression heatmaps were generated using the R package pheatmap (v1.0.12), and GO enrichment

analysis was performed using Metascape[53] (http://metascape.org). Aligned reads with splice junctions were visualized on the IGV browser.

**Statistics**. No statistical methods were used to predetermine sample size. Results were represented as mean ± standard deviation (s.d.) and all experiments have at least three independent biological repeats unless otherwise noted in the figure legends. Differences between groups were examined for statistical significance using unpaired two-tailed Student's $t$ test (for two groups), one-way ANOVA or two-way ANOVA followed by Tukey's multiple comparisons test (for more than two groups) in GraphPad Prism 8.

**Reporting summary**. Further information on research design is available in the Nature Research Reporting Summary linked to this article.

## Data availability
ChIP-seq and RNA-seq data sets generated in this study have been deposited in NCBI's Gene Expression Omnibus (GEO) under accession code GSE135022. All other data supporting the findings of this study are available from the corresponding authors upon reasonable request. Source data are provided with this paper.

## Code availability
Custom scripts used in this study are available from the corresponding authors upon reasonable request. Source data are provided with this paper.

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

## Acknowledgements

We thank BRC Shared Facilities for mouse husbandry support, the Genome Institute of Singapore (GIS)-Next Generation Sequencing Platform for help with high-throughput sequencing, the SIgN Flow Cytometry Core for help with fluorescence-activated cell sorting, the IMCB Central Imaging Facility for assistance with confocal microscopy, and members of the E.G. lab for discussion. I.R.B. and D.R.H. were recipients of the A*STAR Singapore International Pre-Graduate Award (SIPGA). This work was supported by funding from the National Research Foundation, Singapore (NRFF-2015-05 to D.M.M. and NRF-CRP17-2017-06 to E.G.) and the National Medical Research Council, Singapore (NMRC/OFIRG/0032/2017 to E.G. and NMRC/OFYIRG/0019/2016, NMRC/OFIRG/0015/2016, and NMRC/OFIRG/0038/2017 to H.G.).

## Author contributions

Project conceptualization and design: B.Y.H, D.M.M., E.G.; Experimentation: B.Y.H., I.R.B., D.H.P.Q., D.R.H.; Embryo isolation and manipulations: M.K.Y.S.; Polysome fractionation and analysis: L.T.L., H.G.; NGS library preparation: H.W.; Bioinformatics analyses: C-S.F., B.Y.H.; Manuscript writing and editing: B.Y.H., D.M.M., E.G., with input from all authors.

## Competing interests

The authors declare no competing interests.
