## [Peer Review File · Nature Communications]

Reviewers' comments:

Reviewer #1 (Remarks to the Author):

In this article the authors describe and characterise the role of Prdm10 in early murine development, both in vivo and in vitro. They found that homozygous embryos are lethal post implantation and E4.5 embryos demonstrate a growth phenotype. To better understand the function of Prdm10 in early development the authors derived mouse embryonic stem cells (mESCs) from the their conditional Prdm10 allele and found that mutant ESCs have growth defects while the expression of pluripotency markers is only modestly affected, indicating that the gene is essential for normal growth, but not for maintenance of the pluripotent state. The authors identify Prdm10 targets by ChIP Seq and surprisingly find its binding restricted to the promoter region of annotated genes. RNA-Seq following Prdm10 knock out, alongside their ChIPseq, demonstrated a role for Prdm10 in regulating translation. They then focus on one downstream target, Eif3b, a component of the eIF3 complex that is involved in translation initiation. Knockdown of Eif3b results in severe growth defects that are rescued by over expressing a shRNA-resistant Eif3b. Moreover, they show that overexpression of Eif3b in the Prdm10 null cells gives partial rescue. Finally, the authors do polysome profiling in the Prdm10 mutant cells and show a reduction in translation rates.

The notion that translation can regulate cell state in development would be of fundamental interest. However, the idea that a single new transcription factor activates the transcription of translational regulators, is not particularly novel. If Prdm10 had a lineage specific phenotype then this manuscript might have more general interest and in particular, if they could directly link the regulation of translation to that phenotype, this would be an important and exciting observation. The immune fluorescence in figure S1E and F, could suggest a defect specifically in the ICM, but there is no relative quantification. In addition, the phenotype in ESCs is never really explored, while they show relatively little change in Nanog, KLF2, Oct4 and Esrrb there is no real analysis of the phenotype and or differentiation competence of these cells. It is not clear from the current manuscript whether they can passage mutant cells and under what culture conditions they could do this (i.e could they survive in 2i/LIF or other alternative defined culture medias). Based on what is in the paper, this factor could be a general regulator of translation that is expressed in ESCs, explaining why it was pulled out of multiple gene trap screens, but it is hard to see how it is related to development and differentiation.

Specific Comments:

1. The data in this paper is scientifically and technically sound however the authors should improve the characterisation of Prdm10 null mESCs phenotype (as discussed above). The authors conclude that Prdm10 affects cell growth by analysing doubling times, colony formation and morphology. Cell cycle analyses (especially since GO terms associated with cell cycle are also enriched in the differential expression analysis), staining for proliferation markers (like Ki67) would strengthen the claims that Prdm10 is affecting cell growth/proliferation. Additionally, the authors could also investigate cell death (caspase 3 staining) in Prdm10 null cells.
2. The impact of the Prdm10 KO on pluripotency should also be better assessed (as discussed above).

Also details of specific experiments were not clear. In Fig. S2i,j, how many days post-induction was SSEA-1 staining and AP staining done? In Fig. S2j, only one colony is shown for the AP staining, with no quantification. In Fig. S2h, although not significant, the expression of pluripotent genes seems slightly affected by Prdm10 KO. This was three to five days following the Prdm10 knock out, but they only see significant changes in the doubling time of these cells after day five. What is the expression of these genes at the time points where they observe a pronounced phenotype?

3. Although the authors show that Prdm10 leads to a decrease in translation rates that are rescued by Eif3b expression, the Prdm10 phenotype is only partially rescued. What else accounts for the Prdm10 phenotype?

4. Are there other ribosomal proteins whose is regulated by Prdm10 or is it only Eif3b. In particular their GO analysis suggests translation regulation is important, so Prdm10 must be regulating other translation factors.

4. The authors claim that Prdm10 regulates global translation. However, are their mRNA classes that are not affect by Prdm10 KO. If you completely block translation, then one would expect that the cells would not be able to divide at all.

5. In both the embryonic and ESC phenotype, the authors should look at differentiation markers as well as Epiblast.

6. In Prdm10 knock outs (Fig S3A) they still observe Prdm10 ChIP seq peaks, do they remove them from their analysis.

7. In Fig. 2b the label of the Y axis be % of peaks, not peak counts?

8. In Fig. 2E they have a red line for the controls, but never specify what their control is.

9. It is very surprising that they mostly observe binding of this transcription factor at promoters, Why are there no intragenic or enhancer peaks? Is this for a technical reason?

Reviewer #2 (Remarks to the Author):

In this manuscript entitled “Global translation during early development depends on the essential transcription factor PRDM10”, Han et al. document an essential role for PRDM10, an uncharacterized zinc finger containing protein, in pre-implantation embryo and mESCs. PRDM10 apparently functions as a transcriptional activator. Han et al. undertake ChIP-seq and define a motif that responds to the presence of PRDM10. They also define the change in transcriptome 2 and 4 days following ablation of PRDM10. Among the genes whose expression are regulated by PRDM10 is eIF3b. eIF3b is an essential gene so its suppression affects cell survival/proliferation. Ectopic expression of recombinant eIF3b partially rescues loss of PRDM10 (the rescue here is quite small – Figs 4f, g) indicating that other factors are also at play in mediating the PRDM10 lethality. They also show that PRDM10 is essential for normal growth of mESCs but is dispensable for maintenance of the pluripotent state.

The only real weakness that I see in this paper is the link between PRDM10 essentially and effects on eIF3b expression and phenotype. It is not surprising that loss of eIF3b compromises cell growth as the gene has already been defined as essential. The question is how much of PRDM10's effects is a consequence of eIF3b loss versus loss of expression of other targets. In fact, in terms of expression data,

eIF3b ranks ~100th when the list of suppressed genes is rank ordered (Table S3). How many of these other genes, whose expression are more dramatically affected contribute to PRDM10's essentiality? For example, I see eEF1D expression is affected more pronouncedly than eIF3b – why was this not pursued – it is also involved in translation? How many of these 100 mRNAs have PRDM10 binding sites in their regulatory regions? I think the link that PRDM10 regulates eIF3b is credible, the issue is how much of PRDM10's effect is through eIF3b.

Aside from this shortcoming, the paper is well written and the experiments appear to have been well performed. A few additional comments:

p. 7, line 137. “contrast, PRDM10441-880 showed only weak affinity for the MUT probe in direct binding (Fig. 2g) and competition assays (Fig. 2h),”. I don't think claims on affinity can be made here based on these experiments. If such claims are to be made, then K_d 's or K_a 's should be measured.

p. 9, line 176-179. I found the section that reads “Of these, we observed significant differences in expression ($P_{adj} < 0.05$, fold-change > 2) for 52 and 76 genes at 2 and 4 days post-deletion respectively (Fig. 3c). Notably, the majority of genes bound and regulated by PRDM10 ($P_{adj} < 0.05$) showed decreased expression in $Prdm10\Delta/\Delta$ mESCs (Fig. 3d).” confusing. I couldn't understand how many of the genes whose expression are altered in the transcriptome expression data contain PRDM10 binding sites within their promoters. Is it 52 and 76 as stated or 228 – 263 as indicated in Fig 3d.

Please label Y axis on Fig 4H.

What proportion of PRDM10 is nuclear? Cell fractionation or immunofluorescence studies should be able to address this issue.

Rebuttal to reviewers' comments

Reviewer #1 (Remarks to the Author):

In this article the authors describe and characterise the role of Prdm10 in early murine development, both in vivo and in vitro. They found that homozygous embryos are lethal post implantation and E4.5 embryos demonstrate a growth phenotype. To better understand the function of Prdm10 in early development the authors derived mouse embryonic stem cells (mESCs) from the their conditional Prdm10 allele and found that mutant ESCs have growth defects while the expression of pluripotency markers is only modestly affected, indicating that the gene is essential for normal growth, but not for maintenance of the pluripotent state. The authors identify Prdm10 targets by ChIP Seq and surprisingly find its binding restricted to the promoter region of annotated genes. RNA-Seq following Prdm10 knock out, alongside their ChIPseq, demonstrated a role for Prdm10 in regulating translation. They then focus on one downstream target, Eif3b, a component of the eIF3 complex that is involved in translation initiation. Knockdown of Eif3b results in severe growth defects that are rescued by over expressing a shRNA-resistant Eif3b. Moreover, they show that overexpression of Eif3b in the Prdm10 null cells gives partial rescue. Finally, the authors do polysome profiling in the Prdm10 mutant cells and show a reduction in translation rates.

The notion that translation can regulate cell state in development would be of fundamental interest. However, the idea that a single new transcription factor activates the transcription of translational regulators, is not particularly novel. If Prdm10 had a lineage specific phenotype then this manuscript might have more general interest and in particular, if they could directly link the regulation of translation to that phenotype, this would be an important and exciting observation.

We thank the reviewer for summarizing our data. We have revised the manuscript to address some, if not all, the reviewer's concerns as detailed below.

The immune fluorescence in figure S1E and F, could suggest a defect specifically in the ICM, but there is no relative quantification.

What we can conclude from our analysis is that both ICM and TE lineage markers are detectable in the Prdm10 KO embryos, which are delayed in their development, and hence display a difference in morphology.

We indeed agree with the reviewer that the inner cell mass of the embryos is affected by deletion of Prdm10, but a quantification of the *in vivo* results might be confounded by the delay in development of the Prdm10 KO embryos.

Our extensive data presented in the revised manuscript, which follows this initial phenotypic and qualitative assessment of Prdm10 KO ES cells and 8-cell embryos, further suggests that the KO effects, largely mediated by transcriptional loss of eIF3B, do not impact on pluripotent cells specifically.

In addition, the phenotype in ESCs in never really explored, while they show relatively little change in Nanog, KLF2, Oct4 and Esrrb there is no real analysis of the phenotype and or differentiation competence of these cells.

We agree with the reviewer that the analysis of the ESC phenotype in our initial submission was not satisfactory. We have now gone through great length to specifically address this issue and reported our new findings in the revised manuscript. These new data are addressed under specific comments 2 and 6 below.

It is not clear from the current manuscript whether they can passage mutant cells and under what culture conditions they could do this (i.e could they survive in 2i/LIF or other alternative defined culture medias).

We apologize for not being clear in this respect. While mutant cells can still be passaged after recombination, the cell growth is severely impacted and while live cells can still be found 8 days after recombination, the population eventually stops doubling by that point. The importance and essentiality of the transcriptional, regulatory function of PRDM10 is underlined by its requirement both in serum/LIF as well as in 2i/LIF culture conditions. We thank the reviewer for suggesting the alternative culture conditions, as this further strengthens our findings. This new data is now part of our revised manuscript (Suppl Fig 2d). The fact that 2i/LIF culture conditions do not ameliorate the phenotype strengthen our observations that PRDM10 loss does not primarily impact on pluripotency maintenance.

Based on what is in the paper, this factor could be a general regulator of translation that is expressed in ESCs, explaining why it was pulled out of multiple gene trap screens, but it is hard to see how it is related to development and differentiation.

We apologize if our phrasing in our original manuscript misled this reviewer in our interpretation of the phenotype. We did not intend to imply specific PRDM10 effects on mESC differentiation or developmental pathways. The goal of our study is to characterize PRDM10's function in a biologically relevant system – in this case, a phenotype that manifested as early embryonic lethality. The severity of this phenotype and essentiality of PRDM10 merits such detailed investigation, and we believe publication in *Nature Communications*. The use of mESCs in our study serves as a tool to address both biochemical and molecular aspects of PRDM10, studies that are not possible in preimplantation embryos.

We have now taken great care to rephrase respective passages to prevent such interpretations and de-emphasize the “developmental” and “differentiation” aspects accordingly. However, as discussed in the revised text (**page 18**), Prdm10 may very well be important in other developmental contexts, which we are sure will be addressed in future in many studies by the community, but are far beyond the scope of the presented manuscript.

To address the reviewer's concern that PRDM10 “*could be a general regulator of translation*” we are exploring its function in several other contexts, and we observe different phenotypes in a context dependent manner, not consistent with PRDM10 being an essential gene in all examined tissues (despite being expressed ubiquitously). We believe this data belongs to a follow up manuscript, but would like to share some of our results **in Rebuttal Fig.1, for reviewers only**. Using a *Nestin-Cre* deleter strain (**Rebuttal Fig.1a, for reviewers only**), mice are born at the expected Mendelian ratio and *Prdm10* KO littermates are smaller in size (**Rebuttal Fig.1b-c, for reviewers only**). We believe this is due to an overall reduction in GH and Igf1 production in the liver (**Rebuttal Fig.1d, for reviewers only**). The brain, despite being smaller, has a normal morphology, which is very different, for example, from what was observed in *Prmt5* KO mice (**Bezzi et al, G&D 2013**). We hope this data supports the claim that PRDM10 is a TF essential to regulate translation in early embryogenesis, but plays different roles in other tissues, consistent with what observed for other PRDM family members.

Specific Comments:

1. *The data in this paper is scientifically and technically sound however the authors should improve the characterisation of Prdm10 null mESCs phenotype (as discussed above). The authors conclude that Prdm10 affects cell growth by analysing doubling times, colony formation and morphology. Cell cycle analyses (especially since GO terms associated with cell cycle are also enriched in the differential expression analysis), staining for proliferation markers (like Ki67) would strengthen the claims that Prdm10 is affecting cell growth/proliferation. Additionally, the authors could also investigate cell death (caspase 3 staining) in Prdm10 null cells.*

We thank the reviewer for the praise and take great pride in our work. As mentioned above we have now heeded the reviewer's advice to perform a more detailed phenotypical assessment of the *Prdm10* deficient mESCs, focusing mainly on cell proliferation and cell death. While we found that the former was not impacted by the loss of PRDM10 (**Suppl. Fig 2h**), detailed Caspase activity analysis revealed severely elevated program cell death upon PRDM10 deletion, particularly beginning at 5 days post-deletion (new **Fig 1j**). We now describe these new findings in the text. We thank the reviewer for this constructive comment, as we find this highly relevant new data further improves our manuscript.

2. *The impact of the Prdm10 KO on pluripotency should also be better assessed (as discussed above). Also details of specific experiments were not clear. In Fig. S2i, j, how many days post-induction was SSEA-1 staining and AP staining done? In Fig. S2j, only one colony is shown for the AP staining, with no quantification. In Fig. S2h, although not significant, the expression of pluripotent genes seems slightly affected by Prdm10 KO. This was three to five days following the Prdm10 knock out, but they only see significant changes in the doubling time of these cells after day five. What is the expression of these genes at the time points where they observe a pronounced phenotype?*

Again, we find this reviewer's comment and request well-justified and apologize for not being satisfyingly specific in our first version of the manuscript. First, we clarified our existing data both in text and figure legends (**Page 7 and Suppl. Fig.3**). We further repeated experiments to include additional time-points (day 4/6/8 for qPCR, day 4/6 for SSEA-1) and more samples. We also now formally quantified the AP staining data.

As the reviewer rightly pointed out, KO cells seem to show slightly increased expression of some pluripotency genes (eg. *Esrrb*). Given that translation is important for ESC differentiation (Sampath et al. 2008) and that *Prdm10* KO mESCs have global defects in translation (**Fig.5**), we speculate that the decreased translation capacity in KO, leads to preferential killing of cells upon differentiation, skewing the population a more naïve pluripotent state over time.

3. *Although the authors show that Prdm10 leads to a decrease in translation rates that are rescued by Eif3b expression, the Prdm10 phenotype is only partially rescued. What else accounts for the Prdm10 phenotype?*

We do agree with the reviewer and sympathize with his/her curiosity. Naturally PRDM10, as all transcriptional regulators promotes transcription of multiple downstream target genes. In our work we took great care to very accurately document all putative PRDM10-targets, both at the level of DNA binding, transcriptional regulation and by integrating both data sets. In the revised version we went even further and included a complete novel line of experimentation in form of single preimplantation embryo RNA-seq analysis (new **Fig 3b-d**). Our detailed work describes the severe impact of the lost transcriptional activation of eIF3B in PRDM10 KO embryos and mESCs. We are able to phenocopy *in vivo* and *in vitro* the loss of PRDM10 with the loss of eIF3B. Importantly we are able to partially (in terms of cell growth) or fully (in terms of polysome-to-monosome ratio) rescue the observed defects, by re-establishing eIF3B levels in the context of PRDM10 KO cells. Most likely the partial rescue is due to the misregulation of other targets, although we cannot exclude alternative explanations (e.g. eIF3B levels of expression or post-translational modifications are not optimal to ensure a more efficient phenotypic rescue).

Nonetheless, the fact that we can achieve a very significant rescue to the extents shown (**Fig.5**) is in our view already remarkable. We are now, in our revised manuscript, discussing the justification to focus on eIF3B in much detail and also emphasize the fact that other factors may and will contribute to the phenotype (**Page 13–14, Fig 3, Suppl Fig.5**).

In addition, we attempted to address this concern in the revised work by identifying a set of overlapping “PRDM10-bound and regulated genes” in both mESCs and embryos, to increase the likelihood of finding relevant targets for validation. We screened several of these genes by siRNA or shRNA knockdown in embryos as well as mESCs. Most results were negative or inconclusive. *Eif3b* was the strongest hit, hence we prioritized it for validation, but preliminary data pointed to a couple other hits, e.g. *Rpl19* (**Rebuttal Fig 2 for reviewers only**).

In conclusion, we hope we have justified our focus on eIF3B, as testing all bound and regulated candidates to achieve an indeed near-impossible full rescue of the PRDM10 phenotype is a very unsure and unlikely undertaking.

Below (point 4) we discuss additional candidates that may have an impact on the observed phenotype.

4. *Are there other ribosomal proteins whose is regulated by Prdm10 or is it only Eif3b. In particular their GO analysis suggests translation regulation is important, so Prdm10 must be regulating other translation factors.*

Indeed, a justified question and observation by the reviewer. A number of genes involved in translation have altered expression levels, in particular in knock-out mESCs. Many of these are merely regulated, not bound by PRDM10, hence unlikely direct PRDM10-targets. This will be reflected in the GO analysis of regulated genes. We speculate, that this observed misregulation of ribosomal genes originates from the direct impact on *Eif3b*.

That being said, we have identified *Rpl19* (60S Ribosomal Protein L19) as a direct target of PRDM10 that is involved in ribosomal function and translation. *Rpl19* is modestly downregulated in PRDM10-deficient mESCs (~1.6-fold reduction, *P*_{adj} = 4.33E-110 (Day 4); **Supplementary Table S3**). It is also essential for mESC survival, as shown in siRNA and shRNA knockdown experiments (**Rebuttal Fig 2 for reviewers only**). As *Rpl19* depletion is sufficient to inhibit mESC growth, we think it is likely that *Rpl19* downregulation contributes to the Prdm10 KO phenotype in mESCs. However, our embryo RNA-seq data indicated that *Rpl19* expression was not significantly reduced in *Prdm10*-null embryos (*P*_{adj} = 0.097), ranking far below *Eif3b* (*P*_{adj} = 5.76E-09), leading us to prioritize *Eif3b* over *Rpl19* in validation studies. It remains a strong possibility that *Rpl19* is a relevant PRDM10 target in the context of protein synthesis, and we suggest that this could be a suitable topic for follow-up studies.

Another factor we have identified as direct target, (i.e. bound and regulated by PRDM10) is the translation elongation factor *Eef1d*. Our own and new data (now addressed on **Page 14, and Rebuttal Fig 2 for reviewers only as well as Fig.5 and Suppl. Fig.5**), as well as published data indicate that *Eef1d* is not essential in ESCs and therefore we did not pursue this in detail.

5. *The authors claim that Prdm10 regulates global translation. However, are their mRNA classes that are not affect by Prdm10 KO. If you completely block translation, then one would expect that the cells would not be able to divide at all.*

This is a just concern and we apologize for not making our observations clear. We do not believe translation is entirely blocked in Prdm10 KO cells, nor eIF3B is entirely lost. We still observe eIF3B expression, yet at much reduced levels. We also observe a prevailing polysome fraction, albeit also largely reduced. A full block of translation will indeed result in full cell cycle arrest and shown by the treatment of E14 cells with 2.5ug/ml CHX, killing cells within less than 24hs (data not shown). The PRDM10 KO and eIF3B knock-down phenotypes are clearly less severe.

6. *In both the embryonic and ESC phenotype, the authors should look at differentiation markers as well as Epiblast.*

We agree with this request, which is in line with previous comments to further characterize the impact of PRDM10 loss in mESCs. We have gone through great length to characterize differentiation markers and included these findings in our revised manuscript. To address this, we assessed the expression of several well-characterized pluripotency markers at multiple time-points (up to 8 days) after *Prdm10* deletion. Global transcriptome analysis of *Prdm10*^{Δ/Δ} mESCs compared to controls at days 2 and 4 post-deletion showed no significant downregulation of genes associated with mESC pluripotency and self-renewal; in particular, the transcription factors comprising the core pluripotency regulatory circuitry (*Pou5f1*, *Klf4*, *Sox2*, *Nanog*) are expressed at levels comparable to or slightly higher relative to controls (**Supplementary Fig. 3a**). As further validation, we examined selected pluripotency markers (*Nanog*, *Pou5f1*, *Klf2*, *Klf4* and *Esrrb*) by qRT-PCR at day 6 and 8 post-deletion, and confirmed that their expression was maintained even at time-points where *Prdm10*^{Δ/Δ} mESCs exhibit significant growth and survival defects (**Supplementary Fig. 3b**). Notably, these findings mirror our *in vivo* observations that OCT4 and NANOG remain expressed in *Prdm10*^{Δ/Δ} E4.5 embryos (**Supplementary Fig. 1e, f**).

Similarly, we detected no reduction in SSEA-1 surface expression on *Prdm10*-null mESCs at day 4 and 6 post-deletion (**Supplementary Fig. 3c**). *Prdm10*^{Δ/Δ} mESCs formed colonies smaller than that of controls, but nonetheless stained positive for alkaline phosphatase activity and showed a level of AP-positive colony formation ability comparable to that of controls, even at day 7 post-deletion (**Supplementary Fig. 3d**). Further transcriptomic analysis of *Prdm10*^{Δ/Δ} mESCs cultured under SL conditions revealed no significant misregulation of germ layer lineage markers (**Supplementary Fig. 3e**), confirming that loss of PRDM10 does not induce precocious differentiation. Taken together, our results indicate that PRDM10 promotes normal growth of mESCs and early embryos but is dispensable for the maintenance of the pluripotent state. Last, we have performed mESC differentiation into Embryoid bodies and performed lineage marker expression analysis. We show that KO cells are able to form EBs and induce expression of lineage markers (**Rebuttal Fig 3 for reviewers only**).

7. *In Prdm10 knock outs (Fig S3A) they still observe Prdm10 ChIP seq peaks, do they remove them from their analysis.*

Thank you for this observation. We should have clarified this in our original manuscript, which we have now done on page 8: "...all peaks detected in wild-type cells were either absent or strongly diminished in PRDM10-depleted cells (Supplementary Fig. 4a)".

While we do observe peaks in KO cells all peaks are much weaker than their counterparts in WT cells. We make the reasonable assumption that they reflect binding by residual PRDM10 protein at 48 h post-deletion, the time-point at which we harvested cells for ChIP.

This observation rather confirms these peaks as true binding sites, for which ChIP-seq signals are diminished due to PRDM10 protein reduction. Hence, these peaks were not removed from analysis.

8. *In Fig. 2b the label of the Y axis be % of peaks, not peak counts?*

In accordance with ChIPpeakAnno R package used for this analysis, peak counts is the correct labelling for this graph.

9. *In Fig. 2E they have a red line for the controls, but never specify what their control is.*

We apologize for this oversight. We have now clearly stated our intention in the figure legend (**Fig 2e**). Control: all genomic regions within ± 1 kb of gene TSS. Y-axis: phyloP

vertebrate conservation score. Shaded regions: 25%–75% percentile of conservation scores.

10. It is very surprising that they mostly observe binding of this transcription factor at promoters. Why are there no intragenic or enhancer peaks? Is this for a technical reason?

We do not exclusively observe ‘promoter peaks’, which we have now stated in the revised text (“...9.8% (of peaks) mapping to intergenic regions (Fig. 2a)...”). Fig 2a also shows ~10% peaks located in introns, which are intragenic, however, the majority of peaks does map to promoter regions, which we do think is a surprising feature for a transcription factor. We remind the reviewer that the presented peaks are of high-confidence obtained from three individual ChIP-seq experiments using three distinct antibodies.

Reviewer #2 (Remarks to the Author):

In this manuscript entitled “Global translation during early development depends on the essential transcription factor PRDM10”, Han et al. document an essential role for PRDM10, an uncharacterized zinc finger containing protein, in pre-implantation embryo and mESCs. PRDM10 apparently functions as a transcriptional activator. Han et al. undertake ChIP-seq and define a motif that responds to the presence of PRDM10. They also define the change in transcriptome 2 and 4 days following ablation of PRDM10. Among the genes whose expression are regulated by PRDM10 is eIF3b. eIF3b is an essential gene so its suppression affects cell survival/proliferation. Ectopic expression of recombinant eIF3b partially rescues loss of PRDM10 (the rescue here is quite small – Figs 4f, g) indicating that other factors are also at play in mediating the PRDM10 lethality. They also show that PRDM10 is essential for normal growth of mESCs but is dispensable for maintenance of the pluripotent state.

The only real weakness that I see in this paper is the link between PRDM10 essentially and effects on eIF3b expression and phenotype. It is not surprising that loss of eIF3b compromises cell growth as the gene has already been defined as essential. The question is how much of PRDM10's effects is a consequence of eIF3b loss versus loss of expression of other targets. In fact, in terms of expression data, eIF3b ranks ~100th when the list of suppressed genes is rank ordered (Table S3). How many of these other genes, whose expression are more dramatically affected contribute to PRDM10's essentiality? For example, I see eEF1D expression is affected more pronouncedly than eIF3b – why was this not pursued – it is also involved in translation? How many of these 100 mRNAs have PRDM10 binding sites in their regulatory regions? I think the link that PRDM10 regulates eIF3b is credible, the issue is how much of PRDM10's effect is through eIF3b.

First, we'd like to thank this reviewer for his or her praise for our work. We greatly appreciate the positive feedback and take it as motivation to excel and improve the manuscript further. We also can relate to the reviewer's only major concern. In this revised version of our manuscript, we have provided additional data, experiments and reasoning relating to this concern, which has been in part raised by reviewer 1 (comment 4) as well.

'In terms of expression, eIF3B ranks 100th when ordered by expression'. This is indeed true, but for us not of concern. Firstly, the list the reviewer refers to are merely regulated genes, not bound-and-regulated, hence direct target genes.

We now made this distinction clear in the text, and are providing a detailed list of regulated and bound genes, and integration of both datasets. The level of misregulation – i.e. differential expression of KO vs. control – is not necessarily an indication of impact or importance of the gene. As shown in our siRNA or shRNA knock-down experiments, a moderate loss of eIF3B (as seen in the *Prdm10* KO) has a profound phenotypic effect.

We have indeed noted as well that eEF1D is also a misregulated direct target. Our literature research, which we have now depicted in detail in the revised manuscript however indicated that eIF3B rather than eEF1D has essential roles *in vivo*. Indeed, amongst the identified direct targets, eIF3B has the most promising *in vivo* phenotype – causing preimplantation embryonic lethality, phenocopying what we have observed in PRDM10 KO mice.

Most importantly, however, are our new insights *in vivo*. We have now conducted *in vivo* RNA seq experiments on individual E2.5 (8 cell stage control and mutant *Prdm10* KO) embryos. Here, the impact of the *Prdm10* deletion results in a much more concise pool of targets, which are further narrowed down when cross referenced with the ESC chip-seq data. Again, eIF3B is the most promising candidate. Finally, and we believe most irrefutable, we show a fully penetrant phenocopy of the PRDM10 KO effect *in vivo* by specifically knocking down eIF3B by siRNA injection into WT zygotes (new Fig.4). In addition to our *in vitro* phenocopy and partial rescue of growth effects and full rescue of the polysome-to-monosome ratio in mESCs (new Fig.5) we now provide strongest evidence of the PRDM10-eIF3B axis to be main cause for the observed phenotype.

All these new data and discussion are now incorporated in the new manuscript and we are sure that this will satisfy the reviewer's concerns.

Aside from this shortcoming, the paper is well written and the experiments appear to have been well performed.

Thank you, we take pride in our work.

A few additional comments:

p. 7, line 137. "contrast, PRDM10441-880 showed only weak affinity for the MUT probe in direct binding (Fig. 2g) and competition assays (Fig. 2h)," I don't think claims on affinity can be made here based on these experiments. If such claims are to be made, then Kd's or Ka's should be measured.

We absolutely agree with the reviewer's remark and did not mean to claim quantitative measures. We have now revised this passage in the new manuscript and make sure to emphasize a qualitative rather than quantitative measure.

p. 9, line 176-179. I found the section that reads "Of these, we observed significant differences in expression ($P_{adj} < 0.05$, fold-change > 2) for 52 and 76 genes at 2 and 4 days post-deletion respectively (Fig. 3c). Notably, the majority of genes bound and regulated by PRDM10 ($P_{adj} < 0.05$) showed decreased expression in Prdm10 Δ/Δ mESCs (Fig. 3d)." confusing. I couldn't understand how many of the genes whose expression are altered in the transcriptome expression data contain PRDM10 binding sites within their promoters. Is it 52 and 76 as stated or 228 – 263 as indicated in Fig 3d.

We realize that we applied different criteria in specific sub-figures. i.e. bound-and-regulated, regulated significantly and 2-fold, or merely significantly regulated. This is of course cause of confusion and not helpful. We have now used a unifying selection criteria: $P < 0.05$ + fold-change > 2 , to describe differential gene expression avoiding confusion and providing consistent and less confusing numbers. We apologize for overlooking this in our initial manuscript. We have heavily revised these relevant figures and text, by further integrating the *in vivo* RNA-seq data (**Fig.3 and Suppl. Fig.3-5**).

Please label Y axis on Fig 4H.

Thank you for noting this. This figure is now **Fig. 5** in our revised manuscript and the axis label is edited from "A254" to "Abs. (254nm)", abbreviation for "absorbance at 254nm".

What proportion of PRDM10 is nuclear? Cell fractionation or immunofluorescence studies should be able to address this issue.

We thank the reviewer for the comment. PRDM proteins are mostly nuclear and we have strong evidence that PRDM10 binds efficiently to chromatin. Antibody validation data from other sources (eg. <https://www.atlasantibodies.com/products/antibodies/primary-antibodies/triple-a-polyclonals/prdm10-antibody-hpa026997/>), confirm its nuclear localization.

We have tried to perform more detailed experiments by immunofluorescence staining, but unfortunately did not succeed in getting a detectable and reliable signal.

We apologize for not having new data regarding this aspect, but we'll be careful in highlighting only a role of PRDM10 as a transcription regulator in the nucleus. Its potential role outside of the nucleus will be a fascinating topic of investigation for future projects.

APPENDIX FIGURES

Rebuttal Figure 1 (for reviewers only):
Prdm10 is not essential for neurogenesis

Rebuttal Figure 1. *Prdm10* is not essential for neurogenesis.

(a) qRT-PCR analysis of *Prdm10* exon 5 expression in brain tissue isolated from 6-week-old male and female mice of the following genotypes: Nes-Cre⁻ (WT), *Prdm10*^{F/+}; Nes-Cre⁺ (HET), and *Prdm10*^{F/F}; Nes-Cre⁺ (KO).

(b) Representative photos of male littermates at 6 weeks of age, illustrating *Prdm10* gene dosage-dependent size differences between WT, HET and KO mice.

(c) Body weights of male and female *Prdm10* Nes-Cre mice measured at 1 to 6 weeks of age, indicating severely stunted growth in KO mice and moderate effects in HET mice.

(d) qRT-PCR analysis of *Igf1* mRNA in liver tissue from 6-week-old male and female mice of indicated genotypes.

(e) No gross anatomical differences were detected by H&E histological analysis of sagittal brain sections from mutant and control mice.

Expression normalized to *Hprt*, each point represents an individual mouse (**a** and **d**). Data presented as mean ± s.d, **P* < 0.05, ***P* < 0.01, ****P* < 0.001, *****P* < 0.0001; two-tailed unpaired Student's *t*-test.

Rebuttal Figure 2 (for reviewers only): Investigation of other candidate *Prdm10* target genes

Rebuttal Figure 2. Investigation of other candidate *Prdm10* target genes.

(a) Validation of candidate target genes by siRNA knockdown in wild-type embryos cultured to blastocyst stage. Control (n = 38), *Cmc2* (n = 80), *Dut* (n = 44), *Eif3b* (n = 40), *Selenow* (n = 93), *Ss18* (n = 87), and *Ube2a* (n = 66). Y-axis: percentage of cavitated or non-cavitated blastocysts of total embryos analyzed in each experiment, for at least 3 independent experiments per target gene.

(b) Candidate target genes were depleted in E14 mESCs by siRNA transfection, and viable cells measured 72 h post-transfection by CellTiterGlo assay. Data presented as mean \pm s.d., * $P < 0.05$, n.s. not significant; two-tailed unpaired Student's *t*-test. Each point represents 1 replicate sample, representative data shown from 1 out of 2 independent experiments.

(c) Growth curves for E14 mESCs transduced with 2 shRNA constructs targeting *Rpl19* (*Rpl19-391*, *Rpl19-393*), vs. SCR control, at indicated time-points after completion of puromycin selection.

(d) qRT-PCR analysis of *Rpl19* knockdown in shRNA-transduced cells.

(e) Growth curves for E14 mESCs transduced with 2 shRNA constructs targeting *Eef1d* (*Eef1d-194*, *Eef1d-262*), vs. SCR control, at indicated time-points after completion of puromycin selection.

(f) qRT-PCR analysis of *Rpl19* knockdown in shRNA-transduced cells.

Rebuttal Figure 3 (for reviewers only): Differentiation competence of *Prdm10* KO mESCs

Rebuttal Figure 3. Differentiation competence of *Prdm10* KO mESCs.

(a) Experimental set-up for embryoid body differentiation and analysis. Undifferentiated mESCs were dissociated to single-cell suspension by trypsinization, diluted in ES culture medium without mLIF and seeded in 25 μ l hanging drops at a density of 100 or 400 cells per drop for analysis at Day 4 or Day 6 post-induction respectively.

(b) Representative brightfield images of embryoid bodies derived from control and KO mESCs. *Prdm10* KO cells form EBs that initially appear indistinguishable from controls (Day 2), but later on begin to disintegrate, showing morphological features of cell death (Day 4). Scale bar: 100 μ m.

(c) Expression of lineage markers for all three germ layers in EBs harvested at indicated time-points, normalized to *Ubb* and presented as fold-change relative to undifferentiated mESCs (Day 0). Each point represents a biological replicate comprising approximately 30 pooled EBs. RNA was extracted using the Arcturus PicoPure RNA Isolation Kit (Applied Biosystems) and converted to cDNA using the High-Capacity cDNA Reverse Transcription Kit (Applied Biosystems).

Reviewers' comments:

Reviewer #1 (Remarks to the Author):

The authors have done an excellent job at addressing our comments. However, there are still a few issues with the manuscript that should be addressed.

In supplemental Fig 1E, they show the expression of Nanog and Oct4 in the trophoblast of PDRM10 mutants. There also appears no ICM. However, these images would be greatly improved by including an overlay. In addition, the control embryos in this figure do not have a completely normal morphology. Is the ICM missing in the mutants missing and the co-expressing ICM/trophoblast genes - this is what appears in the figure? How does this fit with the ex vivo culture data? Clearly better staining, for both in vivo derived and ex vivo cultured embryos (including the time points shown in bright field), quantification of expression and n values are needed here. They should also augment this with more extensive analysis of their RNA seq from wild type and null embryos.

A similar issue exists with respect to the ESC phenotype. In all cases their control cells show gene expression trends (e.g. Fig 3d, S3A), why does cell culture over 2 days produce these differences? The analysis is quite difficult to interpret as the gene expression changes they see relative to the controls do not always appear as significant as the changes between the time course for the controls. They are also missing the figure legend for S3e.

The differentiation experiments in rebuttal figure 3 should be included, but with additional analysis of pluripotency genes in differentiation (particular given their ectopic expression in vivo).

Rebuttal figure 2 on additional translational related targets should also be included in a revised manuscript.

On line 243 there are two different n values for the same 8.7%. Assume this is just a typo.

Reviewer #2 (Remarks to the Author):

This is a resubmission in which the authors have added new information to bolster their claim that the phenotypic effects of PDRM 10 are through eIF3b. This paper was previously rejected but the authors have decided to come back with new data claiming that this link is better supported.

Figure 4 is completely new and is provided to show that eIF3b is an essential gene in early embryos and mESCs, which is good to know but does not strengthen the PDRM10 relationship.

The panels for new Fig 5 were in the previous MS, so there is no new information here.

At the end of the day, the fact that a transcription factor regulates an essential translation factor, and that translation is affected when you remove the transcription factor is an expected result. The fact that you can't completely rescue the translation defect by OE of eIF3b says something else is at play here and

makes it difficult to get excited about claims of PRDM10 phenotype being "largely mediated through eIF3b-dependent effects on global translation". (ABSTRACT).

Authors' response to reviewer comments

Reviewer #1 (Remarks to the Author):

The authors have done an excellent job at addressing our comments. However, there are still a few issues with the manuscript that should be addressed.

We are delighted that our efforts to further improve the manuscript are recognized by the reviewer as an 'excellent job', and have again made great additional efforts to address the final few issues raised here.

We have identified one major misunderstanding regarding the embryonic phenotype, no doubt due to a lack of clarity on our side, which we address here in much detail:

We very much appreciate the reviewer's interest and curiosity in the PRDM10 embryonic phenotype and possible impact on blastocyst and ICM formation. It appears however that we failed to convey our observations adequately and apologize for this: The matter of fact is that PRDM10 mutant embryos do NOT form blastocysts, which means that these embryos do not display clear or most of the time any ICMs. While we can derive embryos at day 3.5 and even at day 4.5 of development, these do NOT resemble normal embryos, but – as stated in the original version of the manuscript – morula-like, abnormal structures. We observe this at high frequency and have quantified these observations (**Fig.1c** and **Supplementary Fig.1d**).

Again, it is clear to us now that we have failed to communicate our findings properly and appreciate the reviewers' critique to improve this section of the manuscript, which we have now done as follows:

In first instance we have **modified Fig.1**. To this end, we have removed the original panel (b) and replaced it with enlarged representative images of 2 control and 2 mutant embryos at E3.5 of development. As the E4.5 mutant embryos do not show any further developmental progression compared to E3.5 embryos, these panels have been removed as they were merely confusing the reader.

As requested, we have emphasised the quantification of the blastocyst formation capabilities of mutant and control embryos, and added numbers and percentages to **Fig.1c**. This clearly shows the developmental arrest of mutant embryos before the blastocyst stage. It is hopefully now more evident that these embryos do not form proper blastocoels and ICMs.

We have conducted the *in vitro* development experiments to monitor the developmental progression and phenotypic defects in mutants carefully across preimplantation stages. In essence, *in vitro* grown embryos recapitulate the *in vivo* development exactly and embryos arrest at the morula to blastocyst transition. As these experiments add no new data, but much clarity to the story, we have now moved this panel to **Supplementary Fig.1e**. We have improved both image quality and magnification, and added more examples of mutant and control embryos. As requested, we also added the precise timings when these embryos were analysed/photographed. We hope these changes provide more detailed information in the revised manuscript as well as simplify its flow.

It is well established that embryos lacking OCT4 or CDX2 fail to form normal blastocysts. Defective expression of genes driving the first lineage segregation could have been one explanation for the observed PRDM10 KO phenotype. We therefore, early on, made an effort to analyse the expression of the major factors in this lineage segregation, namely OCT4 and CDX2. As both factors are indeed detectable in the KO embryos, we have concluded, and clearly show so in the course of this manuscript, that PRDM10 acts on other

pathways than OCT4/CDX2/NANOG. We have now taken great care to improve this description in the revised version of the manuscript. To this end, we have repeated IF analyses and now show more representative control and mutant embryos for both OCT4/CDX2 and NANOG expression analysis in **Supplementary Fig.2a–b**. Co-expression of OCT4 and CDX2 in morulae at early stages of lineage segregation are well documented in the literature (e.g. Dietrich and Hiiragi, Development 2007). Even at the blastocyst stage, occasional OCT4/CDX2 double-positive TE cells can be observed (**Supplementary Fig.2a**, indicated by arrowheads). As PRDM10 mutant embryos arrest at the morula to blastocyst transition, it is therefore not surprising that such co-expression is prevalent. This would in fact be expected. Furthermore, these embryos are arresting and dying. Apoptotic cells are for instance apparent by their pyknotic nuclei in the DAPI stain (**Supplementary Fig.2a**, indicated by asterisks).

Because PRDM10 mutant E3.5 embryos are so phenotypically abnormal, we chose to address gene expression by RNA-seq at the 8-cell stage when embryos are still intact and normal in appearance (see *in vitro* culture, **Supplementary Fig.1e**). As requested, we have now used this data to address and quantify the expression levels of lineage segregation markers, and do not find significant differences between mutant and controls (**Supplementary Fig.2c**).

In essence, PRDM10 KO embryos arrest before blastocyst formation and therefore fail to segregate ICM and TE; however, they are able to express both markers and thus unlikely to arrest due to lineage segregation block. Instead, we show compelling evidence that these embryos are highly defective in translation initiation.

We hope these changes and explanations clarify the misconceptions about the phenotypic impact of PRDM10 loss. Below are our detailed responses to the reviewer's queries:

In supplemental Fig 1E, they show the expression of Nanog and Oct4 in the trophoblast of PDRM10 mutants. There also appears no ICM. However, these images would be greatly improved by including an overlay.

We agree that in mutant embryos OCT4 and CDX2 are co-expressed. This is normal and expected for morula stages at which these KO embryos are arrested. We have now included overlaps of both CDX2/OCT4 alone and CDX2/OCT4 and DAPI (DNA), which indeed improved these figures substantially. We also have included data from additional, new embryos for further clarity.

In addition, the control embryos in this figure do not have a completely normal morphology. Is the ICM missing in the mutants missing and the co-expressing ICM/trophoblast genes - this is what appears in the figure?

Again, we have replaced the images of control embryos and added more details. Mutant embryos do not form ICMs and co-express TE and ICM genes, as expected at the morula stage.

How does this fit with the ex vivo culture data? Clearly better staining, for both in vivo derived and ex vivo cultured embryos (including the time points shown in bright field), quantification of expression and n values are needed here.

As mentioned above, the *ex vivo* culture underlines the *in vivo* observation of embryonic arrest at the morula stage. We have now expanded these panels (**Supplementary Fig.1e**) with more controls and mutants and added the timepoints shown in the Figure.

They should also augment this with more extensive analysis of their RNA seq from wild type and null embryos.

In the revised **Supplementary Fig.2c**, we present a new analysis of RNA-seq data from 8-cell stage embryos to show that there are no changes in ICM and TE lineage marker expression between PRDM10-null and control embryos, consistent with our immunofluorescence staining data (**Supplementary Fig.2a–b**).

A similar issue exists with respect to the ESC phenotype. In all cases their control cells show gene expression trends (e.g. Fig 3d, S3A), why does cell culture over 2 days produce these differences? The analysis is quite difficult to interpret as the gene expression changes they see relative to the controls do not always appear as significant as the changes between the time course for the controls.

We thank the reviewer for this comment and wish to clarify that it was not our intention to make any claims regarding temporal changes in gene expression over a time-course, but only to compare gene expression differences in PRDM10 KO mESCs relative to controls *within each time-point*. The apparent “differences” in the original figure were likely due to technical reasons related to sample processing, and not indicative of a true biological difference. Consistent with this, RT-qPCR data from an independent set of experiments (**Fig.5c**) show that *Eif3b* expression in control cells is unchanged between Day 4 and Day 2, and we have observed the same for other target genes as well (data not shown).

We apologize for the confusion, and have reformatted the heatmaps in **Fig.3d**, **Supplementary Figs. 4a, 4e** with gene expression values normalised separately for Day 2 and Day 4 to represent relative changes between KO and controls within each time-point, for a more accurate representation of the data. However, we emphasise that this does not change the results of our analysis or affect the validity of our main conclusions.

They are also missing the figure legend for S3e.

We thank the reviewer for pointing this out. It was omitted by mistake, and is now included in the revised manuscript.

The differentiation experiments in rebuttal figure 3 should be included, but with additional analysis of pluripotency genes in differentiation (particular given their ectopic expression in vivo).

As requested by the reviewer, the EB differentiation data from Rebuttal Figure 3 is now included in the revised manuscript, with the further addition of new RT-qPCR data on pluripotency marker expression in PRDM10 KO vs. control EBs (**Supplementary Fig.5**).

Rebuttal figure 2 on additional translational related targets should also be included in a revised manuscript.

As requested, these data have been integrated into the revised manuscript (**Supplementary Fig.8**), with the inclusion of an additional experimental repeat for the *Eef1d* and *Rpl19* shRNA experiments (**Supplementary Fig.8c–g**).

On line 243 there are two different n values for the same 8.7%. Assume this is just a typo.

We apologise for the typo and thank the reviewer for noticing this. The numbers are 8.2% upregulated and 8.7% downregulated genes, and have now been corrected in the revised manuscript.

Reviewer #2 (Remarks to the Author): *This is a resubmission in which the authors have added new information to bolster their claim that the phenotypic effects of PRDM10 are through eIF3b. This paper was previously rejected but the authors have decided to come back with new data claiming that this link is better supported.*

Figure 4 is completely new and is provided to show that eIF3b is an essential gene in early embryos and mESCs, which is good to know but does not strengthen the PRDM10 relationship.

The panels for new Fig 5 were in the previous MS, so there is no new information here. At the end of the day, the fact that a transcription factor regulates an essential translation factor, and that translation is affected when you remove the transcription factor is an expected result. The fact that you can't completely rescue the translation defect by OE of eIF3b says something else is at play here and makes it difficult to get excited about claims of PRDM10 phenotype being "largely mediated though eIF3b-dependent effects on global translation".

We agree that the incomplete rescue of the PRDM10 phenotype by EIF3B overexpression points to additional contributions by other target genes. We have addressed this point in the revised manuscript, and included new data (**Supplementary Fig.8**, previously in rebuttal) to address the potential role of two other PRDM10 target genes (*Eef1d*, *Rpl19*) that are known to be involved in protein translation.

Accordingly, we have toned down the abstract to make the more conservative claim that the PRDM10 phenotype is "in part mediated through EIF3B-dependent effects on global translation".

REVIEWERS' COMMENTS:

Reviewer #1 (Remarks to the Author):

The authors have addressed the comments we raised.